

**Estimation of duration and its changes in Lagrangian observations relying on ice floes in**
**the Arctic Ocean utilizing sea ice motion product**
Fanyi Zhang[1,2], Ruibo Lei[1,2*], Meng Qu[2], Na Li[2], Ying Chen[2], Xiaoping Pang[1*]
[1]Chinese Antarctic Center of Surveying and Mapping, Wuhan University, Wuhan 430079, China
[2]Key Laboratory for Polar Science of the MNR, Polar Research Institute of China, Shanghai 200136, China
*Correspondence to:* leiruibo@pric.org.cn & pxp@whu.edu.cn
**Abstract:** Since the 1890s, buoy- and camp-based Lagrangian observations relying on ice floes, have been pivotal for
data acquisition during winter in the central Arctic Ocean due to the inaccessibility of most research vessels. Evaluating the
observation duration and its changes associated with changes in Arctic climate system, is crucial for the planning of ice
camp/buoy deployment. Using remote sensing sea ice motion product, we reconstructed sea ice drift trajectories for each
year in 1979–2020 and identified ideal deployment areas of ice camp/buoy in the central Arctic Ocean. The results show that,
based on the setup time of October 1, the areas centered at 82°N and 160°E near the north of East Siberian and Laptev seas,
with a size of $7.6×10^5$ km$^2$, could ensure Lagrangian observations for at least 9 months with the drifting maintaining in the
ice zone and not entering the exclusive economic zones (EEZs) of Arctic coastal countries, with the probability of
76.2%–92.9% during 42 years. The potential deployment areas favored ice advection to the Transpolar Drift (TPD) region
relative to the Beaufort Gyre (BG) region. Ice trajectory endpoints did not reveal an obvious long-term tendency, but were
regulated by large-scale atmospheric circulation patterns, especially the atmospheric patterns in the early drifting stage of
autumn (OND). In particular, the autumn east-west surface air pressure gradient across the central Arctic and the Arctic
Dipole Anomaly indices significantly influenced endpoints of ice trajectories after 9 months and can expand ideal
Lagrangian observation areas under scenarios with their extreme positive phases. The increasing rate of near-surface air
temperatures from autumn to spring along the trajectories was more pronounced in the TPD region than that in the BG
region. The sea ice response to wind stress significantly intensified in recent Lagrangian observations, suggesting stronger
dynamic processes as the sea ice thinning. Geopolitical boundaries of EEZs have a significant impact on the sustainability of
the Lagrangian observations, making it rarely exceed 10 months. Without this restriction, the potential Lagrangian
observations in the BG and TPD regions would expand southward.
**KEYWORDS**: Arctic Ocean; Sea ice; Lagrangian observation; Buoy; Ice camp; Transpolar Drift; Beaufort Gyre



## 1. Introduction

Arctic sea ice, a crucial indicator and amplifier for climate change (Kwok, 2018), has experienced pronounced to become progressively thinner and younger since 1979, with its extent in September declining by 13% per decade during the satellite observation era since 1979 (Parkinson and DiGirolamo, 2021; Meier and Stroeve, 2022; Babb et al., 2023). The state-of-the-art earth system models still have an obvious spread to project the evolution of Arctic sea ice (Notz, 2012), mainly due to insufficient observational data for parameterization of crucial sea ice thermodynamic and dynamic processes (Smith et al., 2022), a severe absence of reliable observation data available for assimilation (Liu et al., 2019), and the rough treatment of Arctic snow and sea ice processes by atmospheric reanalysis data (Batrak and Müller, 2019). The frozen ocean and extremely harsh weather limit the accessibility of the central Arctic Ocean, exacerbating data scarcity of ship-based oceanography measurements. This situation is even worse in the freezing season (Rabe et al., 2022). Lagrangian measurements based on ice camp or buoy deployed on ice floes provide an alternative for the observations of interactions among atmosphere, ice, and ocean in the Arctic. Due to the thicker and more stable sea ice, ice camp or buoy is easier to deploy and maintain in the central Arctic Ocean than in the Southern Ocean, at least until now, which is still in this state.

In the 1890s, Fridtjof Nansen and his companions pioneered Lagrangian observations in the central Arctic Ocean using the ice camp and wooden galleon, which finally provided the first basic depiction of the Arctic sea ice and oceanic physical regimes. Subsequent ice-camp-based campaigns, including the Ice Station Alpha (Cabaniss et al., 1965), the Arctic Ice Dynamics Joint Experiment (AIDJEX; Coon, 1980), the Surface Heat Budget of the Arctic Ocean (SHEBA) cruise (Uttal et al., 2002), as well as the Norwegian young sea ICE (N-ICE2015) Expedition (Granskog et al., 2016), provided vital observation data for the construction of the theoretical framework of sea ice physics, as well as the parameterizations of sea ice thermodynamic and dynamic processes, and heat and/or salt exchanges with lower atmosphere and upper ocean, promoting the developing of sea ice numerical models. The Soviet Union-Russia Arctic ice-camp project, lasting for several decades since the 1930s, have provided extensive climatological characteristics of snow and sea ice geophysical variables of the central Arctic Ocean (Frolov, 2005), supporting numerical simulations (e.g., Tian et al., 2024) and retrieval algorithms of Arctic sea ice (e.g., Lavergne et al., 2010). Recently, the Multidisciplinary drifting Observatory for the Study of the Arctic Climate (MOSAiC), fully leverages the advantages of multidisciplinary observations on the ice floes as an intermediate medium (Nicolaus et al., 2022; Rabe et al., 2022; Shupe et al., 2022), marking a milestone for Arctic drifting observation campaigns.

However, the implementation of ice camp, accompanied by a modern icebreaker as the MOSAiC, requires a significant logistical budget; or without the icebreaker supporting, as the Soviet Union-Russia ice camps, faces with risks including



those from ice floe fragmentation, storms, and polar bears. These factors all limit the sustainable implementation of ice camp.
It is gratifying that Arctic ice floes also provide a broad platform without the need for extra floating support for deploying
buoys or other observation instruments. Various types of buoys are designed and deployed in the Arctic Ocean to measure
sea ice kinematics (Lukovich et al., 2011), snow and sea ice mass balance processes (Richter-Menge et al., 2006; Jackson et
al., 2013; Nicolaus et al., 2021), meteorological parameters and heat exchanges over ice surface (Cox et al., 2023), as well as
oceanic temperature and salinity profiles or turbulence heat flux underneath the ice (Shaw et al., 2008; Toole et al., 2011).
Various types of buoys can also be co-deployed on the same floe to obtain comprehensive observation matrix of multiple
media (e.g., Morison et al., 2002), or at a local scale of tens of kilometers (e.g., Rabe et al., 2024) in order to match the grid
scales of satellite remote sensing (e.g., Koo et al., 2021) and numerical models (e.g., Pithan et al., 2023). This task is
extremely hard to achieve in open water.

The Arctic sea ice is mainly driven by wind and oceanic current stresses, Coriolis force, horizontal gradient force of sea

level, and ice internal stress (Lep[ä]ranta, 2011). Since the complex advection patterns of Arctic sea ice, majorly regulated by
two surface ocean circulation systems of the Beaufort Gyre (BG) and Transpolar Drift (TPD) (Kwok et al., 2013), the setup
or deployment location of ice camp or buoy is considered to be an important factor that determines the effective duration of
observation experiments and the observation regions that may be involved in the subsequent Lagrangian drifting. Remote
sensing sea ice motion (SIM) products can be used to simulate forward (backward) sea ice drift trajectories to track the
destinations (origins) of sea ice (Lei et al., 2019) or estimate ice age by tracking the duration of ice drifting (Tschudi et al.,
2020). Therefore, the main motivation of this study is to identify the ideal deployment locations in the central Arctic Ocean
for ice camp or buoy using SIM product, to ensure that Lagrangian observations can last a sufficiently long period. This is
essential to avoid interruption of observations due to the breakup or collapse of the ice camp or buoy and its supporting ice
floe, and the drifting to the ice edge, or to the exclusive economic zones (EEZs) of one country that is not involved in
observation experiments.

The atmospheric forcing and kinematic mechanism of sea ice during the Lagrangian observations not only determine

the seasonal evolution of sea ice itself, but also affect the energy and momentum exchanges between the atmosphere and sea
ice. They can provide important backgrounds supporting the interdisciplinary studies based on Lagrangian observational data
(e.g., Krumpen et al., 2021; Rinke et al., 2021). Therefore, before planning the deployments of ice camp or buoy, it is also
scientifically valuable to obtain such knowledge of the climatological characteristics and long-term trends of atmospheric
forcing and sea ice kinematics along the subsequent potential drifting trajectory under the background of Arctic amplification
and sustained loss of Arctic sea ice.



Arctic sea ice circulation is generally regulated by atmospheric circulation patterns, such as the Arctic Oscillation (AO),
Dipole Anomaly (DA), Central Arctic air pressure-gradient Index (CAI), and Beaufort High (BH). The AO (Thompson and
Wallace, 1998) regulates the axis alignment of the TPD and the extent of BG. At positive (negative) AO phases, the axis
alignment of the TPD tends to shift westward (eastward) and the BG shrinks (expands) (Rigor et al., 2002). The wind
anomalies induced by DA (Wu et al., 2006) exhibit strong meridional forcing in the TPD region, with positive (negative)
phases accelerating (decelerating) the sea ice drift along TPD (Wang et al., 2009). The CAI, defined as the east-west gradient
of sea level air pressure (SLP) across the central Arctic Ocean could regulate partly meridional wind forcing parallel to TPD
(Vihma et al., 2012). The BH (Moore et al., 2018) is closely associated with sea ice circulation in the BG region
(Proshutinsky and Johnson, 1997). Atmospheric circulation patterns affect sea ice drift trajectory and advection direction
through various mechanisms and consequently affect the duration of Lagrangian observations on the ice floes. Thus, their
regulatory mechanisms and seasonal variations needs further clarification regarding the evaluation of duration of the
Lagrangian observations relying on Arctic ice floes.
In this study, we organized the sections as follows. The datasets and methods used to reconstruct the sea ice drift
trajectory and estimate the changes in atmospheric and ice conditions along the trajectory are briefly described in Sect. 2.
The ideal deployment areas of Lagrangian observations, as well as changes in the atmospheric forcing and ice dynamic
response to wind forcing along the potential ice trajectories during 1979–2020 are presented in Sect. 3. The performance of
the reconstructed method, the connection with the atmospheric circulation patterns of the ice trajectories, and the impact of
EEZ boundary and deployment time on the sustainability of Lagrangian observations are discussed in Sect. 4. Conclusions
are given in the last section. This study provides important supporting information for the planning and implementation of
Lagrangian observations relying on ice floes in the central Arctic Ocean.
**2. Data and methods**
**2.1 Study area**
Our study focuses on the reconstruction of sea ice drift trajectory in the central Arctic Ocean. Here, the central Arctic
Ocean is defined as the high Arctic that excluded from the EEZs of any country, using the maritime boundary polylines
(version 12) of the geodatabase provided by the Flanders Marine Institute. To define the potential areas for identifying
preferred deployment sites, we identified a rectangular area of $1.44\times10^6$ km$^2$, consisting of 2294 pixels on the 25-km
Equal-Area Scalable Earth Grid (EASE-Grid), with area corners aligned with the EEZ boundary polylines, which covers
approximately 51.3% of the central Arctic Ocean we defined (Fig. 1). Although our study region (rectangular area) does not



cover the entire central Arctic Ocean, in order to save computational time, we believe that we do not miss the practicable
area for ice camp or buoy deployment. The reasons for this diagnosis will be given later. Based on the mean Arctic SIM field
in 1979–2020, we roughly defined boundaries to separate the BG and TPD regions, as shown in Fig. 1.

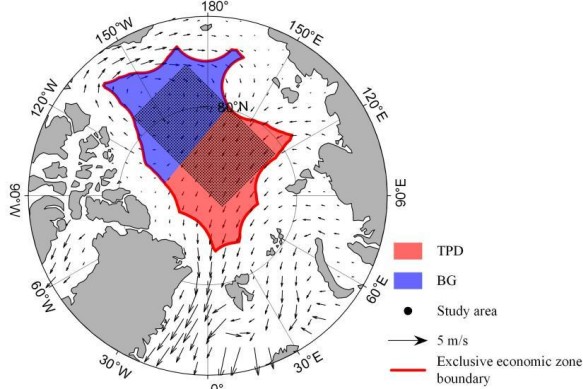


**Figure 1.** Study area. The black dots indicate grid points defined to identify the most optimal area for the buoy or camp deployment. The
arrows depict the mean SIM vectors from 1979 to 2020. The region delineated by the red lines represents the central Arctic Ocean, which
is defined as the high Arctic that excluded the EEZs. The shaded blue and red areas roughly denote the Beaufort Gyre and Transport Drift
regions within the central Arctic Ocean.

**2.2 Data**
**a. Sea ice data**
Due to the difficulty of obtaining long-time series, large-coverage SIM fields from high-resolution remote sensing
images (e.g., Li et al., 2022; Fang et al., 2023), we used the 25-km Polar Pathfinder version 4.1 Sea Ice Motion Vectors from
the U.S. National Snow and Ice Data Center (NSIDC; Tschudi et al., 2020) to reconstruct sea ice drift trajectories originating
from the study area in 1979–2020. The Global Sea Ice Concentration Climate Data Records from the European Organization
for the Exploitation of Meteorological Satellites Ocean and Sea Ice Satellite Application Facility (EUMETSAT OSI SAF;
Lavergne et al., 2019) is utilized for evaluating ice conditions along the trajectory. This sea ice concentration (SIC) data is
derived from the Scanning Multichannel Microwave Radiometer (SMMR), Special Sensor Microwave Imager (SSM/I), and
Special Sensor Microwave Imager/Sounder (SSMIS) passive microwave satellite series sensors. The SIM and SIC data are
projected onto the 25-km EASE-Grid. Sea ice thickness (SIT) along the trajectory is estimated with the merged CryoSat-2
and Soil Moisture and Ocean Salinity (SMOS) observations, hereinafter referred to as CryoSat-2/SMOS (Ricker et al.,
2017b). This dataset, also on a 25-km EASE-Grid, provides weekly SIT data of the freezing season from October through
mid-April since 2010.



**b. Buoy data**

The trajectories of the buoys deployed over the Arctic ice were utilized to validate the reconstructed ice trajectories. To ensure the quality of validated SIM product, we constrained buoy selection to those situated 100 km offshore within the Arctic Ocean and excluded buoys south of the Fram Strait. These buoys were deployed during the German Arctic Research Expedition and the Chinese National Arctic Research Expedition (CHINARE) during the summers of 2014, 2016, and 2018. Details of the buoys are given in Table A1.

**c. Atmospheric data**

Atmospheric conditions were examined using atmospheric reanalysis data from the European Centre for Medium-Range Weather Forecasts Reanalysis v5 (ERA5; Hersbach et al., 2020). Hourly near- surface (2 m) air temperature, 10-m wind and surface longwave radiation at about 30-km horizontal resolution are bilinearly interpolated to derive daily atmospheric conditions along the trajectories.

Seasonal (autumn-OND, winter-JFM and spring-AMJ) atmospheric circulation indices including AO, DA, CAI, and BH were used to characterize the regulatory mechanism of atmospheric circulation patterns on ice trajectories. The AO and DA indices were calculated from the first and second empirical orthogonal functions of the SLP anomalies north of 70°N, utilizing monthly SLP from the National Centre for Environmental Prediction/National Centre for Atmospheric Research to maintain consistency with previous studies (Wu et al., 2006; Wang et al., 2009). Hourly SLP from ERA5 reanalysis was used to calculate the monthly CAI (Vihma et al., 2012), defined as the difference between SLPs at 90°W, 84°N, and 90°E, 84°N. According to Moore et al., (2018), the ERA5 SLP data in the region of 75°–85° N and 170°E–150°W were utilized to define the BH index, which is more compatible with the BG from the perspective of sea ice circulation.

**2.3 Methods**

To assess the effective Lagrangian observation time, a survival time (ST) threshold for floes still drifting within the Arctic ice region and avoiding entering EEZs is crucial. Based on a given ST threshold, regular grids (Fig. 1) were established as the starting point for ice trajectory reconstruction to identify the preferred potential deployment areas of ice buoy or camp. Reconstructed ice trajectories from these areas start on October 1, aligning with the approximate onset of ice freezing season (Markus et al., 2009) and the setup time (October 3) of MOSAiC ice camp (Nicolaus et al., 2022). According to Lei et al., (2019), the ice drift trajectories were reconstructed as follows:

$$X(t) = X(t-1) + U(t-1) \cdot \delta_t, \tag{2}$$



and $Y(t) = Y(t-1) + V(t-1) \cdot \delta_t,$      (3)
where $X$ and $Y$ are the zonal and meridional coordinates of ice trajectories, $U(t)$ and $V(t)$ are the ice motion components at
the time $t$ along the ice trajectories, and the $\delta_t$ is the calculation time step of one day.
When ice floes enter region with SIC < 15% or the EEZ of one country, the reconstructed ice trajectory is terminated,
and the time from October 1 to the terminal point is defined as the ST of ice floe, corresponding to the effective working
duration of ice camp or buoy that is deployed on it. Note that, because the main purpose of this study is to identify and
eliminate areas that are not suitable for deploying ice buoy or camp, with the ST of reconstructed trajectory not meet the
threshold, thereby the truncation of reconstructed ice trajectory in this study is set as one year until 30 September of
following year. As shown in Fig. 2a, with a 10-month ST threshold, the available deployment areas in the central Arctic
Ocean are very limited (22.6% of the rectangular study region), which is much less than the 53.2% area when the ST
threshold is 9 months. The probability with a relative short duration less than 180 days was 1.6% for the effective region
corresponding to the ST threshold of 6 months, which was reduced to a negligible value of 0.5% (0.8%) for the 10-month
(9-month, not shown) ST threshold; while that with a relative long duration of 365 days or beyond was 72.5% for the ST
threshold of 6 months, which increased to 83.7% (79.0%) for the 10-month (9-month, not shown) ST threshold (Fig. 2b).
Therefore, to ensure a broad range of deployment areas, i.e., with a probability of > 50% across the entire study region, and
ensure sufficient duration for Lagrangian observations, we used a 9-month ST threshold for the subsequent analyses.

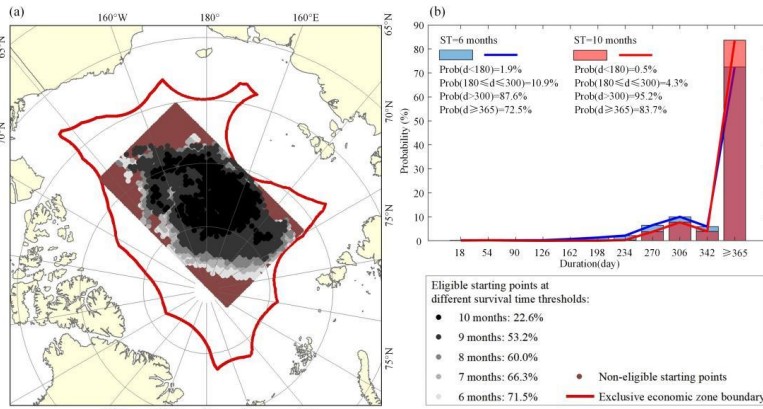


**Figure 2.** (a) Spatial distribution of eligible starting points and (b) probability distribution of duration according to different thresholds of
survival time in 1979–2020. Note that the truncation of reconstructed ice trajectory in this study is one year until 30 September next year.
Thus, the proportions with the duration ≥365 days means the ice trajectory is still valid by 30 September of following year.

To verify the reliability of reconstructed ice trajectories, the Euclidean distance and cosine similarity against the buoy
observations are used to quantify their distance and direction deviations. The Euclidean distance ($D_f$) is defined as follows:





$$D_f = \sqrt{(x_{buoy} - x_{cal})^2 + (y_{buoy} - y_{cal})^2}. \qquad (3)$$
Cosine similarity is an effective metric for assessing the geometrical similarity between the reconstructed trajectories
and buoy trajectories, with a value approaching one denoting a high similarity between them. The cosine similarity ($S_c$)
between the coordinate vectors of the reconstructed trajectory ($\overrightarrow{Q_{cal}}$) and buoy measurement ($\overrightarrow{Q_{buoy}}$) is calculated as follows:
$$S_c = \frac{\overrightarrow{Q_{cal}} \cdot \overrightarrow{Q_{buoy}}}{\left\|\overrightarrow{Q_{cal}}\right\| \left\|\overrightarrow{Q_{buoy}}\right\|}, \qquad (4)$$
where $\overrightarrow{Q_{cal}} = (x_{cal}(i), y_{cal}(i))(i = 1,2,3......)$ and $\overrightarrow{Q_{buoy}} = (x_{buoy}(i), y_{buoy}(i))(i = 1,2,3......)$.
To characterize regional differences between BG and TPD regions, we defined the starting points using the geometric
centers of grid points with a probability of reaching the BG or TPD region greater than 90% (BG: 81.04°N 160.10°W; TPD:
83.43°N 154.75°E) and that having ambiguous destination with a probability of reaching both regions between 40% and
60% (Both: 81.21°N, 175.97°E), and reconstructed ice trajectory over 9 months for each year of 1979–2020 (Fig. 3). The
atmospheric thermodynamic forcing, including the freezing degree days (FDDs) and thawing degree days (TDDs), closely
related to the ice thermodynamic growth and melting processes (Ricker et al., 2017a), as well the surface net longwave
radiative flux, related to the feedback of clouds and sea ice itself on the near-surface atmosphere (Graham et al., 2017), are
estimated along the reconstructed ice trajectories. FDD (TDD) refers to the integral of near-surface air temperatures below
−1.8°C (above 0°C) over the study period. The dynamic response parameters of sea ice to atmospheric forcing are
characterized using the ice-wind speed ratio (Herman and Glowacki, 2012).

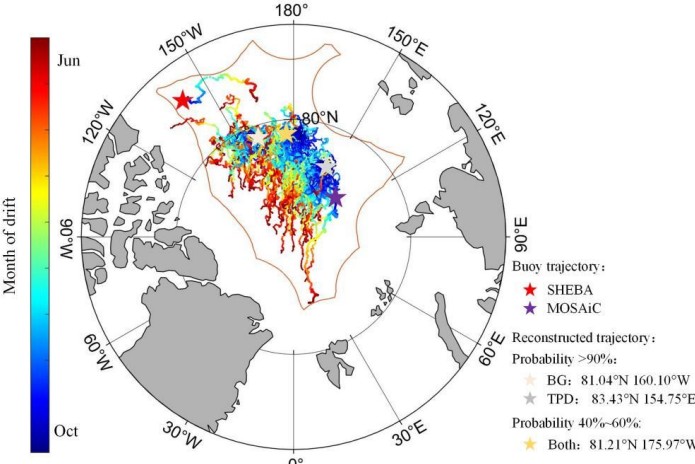


**Figure 3**. Reconstructed 9-month sea ice trajectories for 1979–2020, starting from three geometric centers. For comparison, the partial
drifting trajectories of SHEBA and MOSAiC ice camps started from October 3 of 1997 and 2019 to the time of 9 months after deployment





are also shown.
**3. Results**
**3.1 Spatial distribution of the effective starting points of reconstructed ice trajectories with 9-month ST**
Using the reconstructed ice trajectories for each ice season from 1979 to 2020, the influence of the specific starting
point on the ST and its destination is assessed here. The results reveal that the effective probabilities (the ratio between the
effective years to all study years from 1979 to 2020) of starting points with the reconstructed ice trajectories having
sufficient ST of no less than 9 months ranged from 12.0% to 92.9%. Drifting from the region centered at about 82°N and
160°E, close to the north of East Siberian and Laptev seas, the probability over the 42 years is relatively high than other
regions, generally exceeding 75%. The likelihood of sea ice drifting into the EEZs or beyond the ice zone increased when the
starting point approached the corners of rectangular study region, particularly in the downstream region of the TPD, where
the probability is notably less than 20.0%. This also suggests the rationality of the rectangular study region we defined from
the perspective of identifying the optimal deployment area for ice camp or buoy. The black points shown in Fig. 4b indicated
that the locations (with a size of $7.6 \times 10^5$ km$^2$) as the starting point of reconstructed trajectories with the ST >= 9 months
within 32 years (or 75%) or beyond from 1979 to 2020. This region can be considered as the most ideal area in the central
Arctic Ocean for deploying ice camp or buoy to implement Lagrangian observations.
The probabilities of termination of reconstructed ice trajectory reaching the BG or TPD region during the study period
are illustrated in Fig. 4c–d. Between 1979 and 2020, as expected, the ice floes that tend to drift to the BG region (Fig. 4c) are
mainly originating from the southwest part of the study region; while the ice floes that tend to drift to the TPD region (Fig.
4d) are mainly originating from the northeast part of the study region. However, there is also a large overlapping area
between these two regions, and the magnitude of the probabilities exhibits a regular regional variability pattern for the
specific regions. This suggests that the location of the starting point has a crucial influence on the subsequent ice advection,
or, in other words, the deployment areas of the ice camp or buoy would determine their drift trajectory and final destination
to a high degree. The number of eligible starting points, whose reconstructed trajectory reached the TPD region with ST of
no less than 9 months, accumulated over 75% years of the 42-year study period, was 2.1 times that of such starting points
that reached the BG region. This indicates that sea ice originating from eligible starting points is more likely to reach the
TPD region. For the ice floes originating from the junction zone between the BG and TPD regions (yellow strip in Fig. 4b),
defined using the climatological SIM field, the probability of reconstructed ice trajectories reaching these two regions ranges
from 41.0% to 53.8%, without obvious regional tendency for ice advection destination.



Noting the symbolic shift in the physical nature of Arctic sea ice after 2007 (Sumata et al., 2023), we further calculated
the probability distribution of the starting point with the termination of the reconstructed ice trajectory reaching the BG or
TPD region for the sub periods before and after 2007, as shown in Fig. 5. The probabilities of starting points having the
sufficient ST of no less than 9 months ranged between 14.3% and 92.9% in 1979–2006, which changed to 0.7%–92.9% since
then, indicating a greater variability after 2007 (Fig. 5a–b). The size of ideal deployment area, with a probability > 75% as
shown by black points in Fig. 5c–d, was reduced obviously in 2007–2020 ($4.4×10^5$ km$^2$) by 60.5% compared to that derived
from 1979–2006 ($11.2×10^5$ km$^2$). Such a conspicuous reduce in the preferred area suggests that the deployments of ice camp
or buoy in the Arctic Ocean become more challenging as sea ice decreases.
The spatial distributions of the probabilities of reaching the BG and TPD regions in two sub periods prior to or after
2007 are similar to those derived from the whole study period (Fig. 5e-h); and the spatial proportions of starting points
with > 75% probability of reaching the two regions varied slightly for two sub periods, with changes ranging from 0.9% to
5.1% relative to the full period. This suggests the destination of ice floe advection is relatively stable, which is mainly
associated with the Arctic sea ice circulation patterns (Detailed analysis will be provided in Section 4.2).

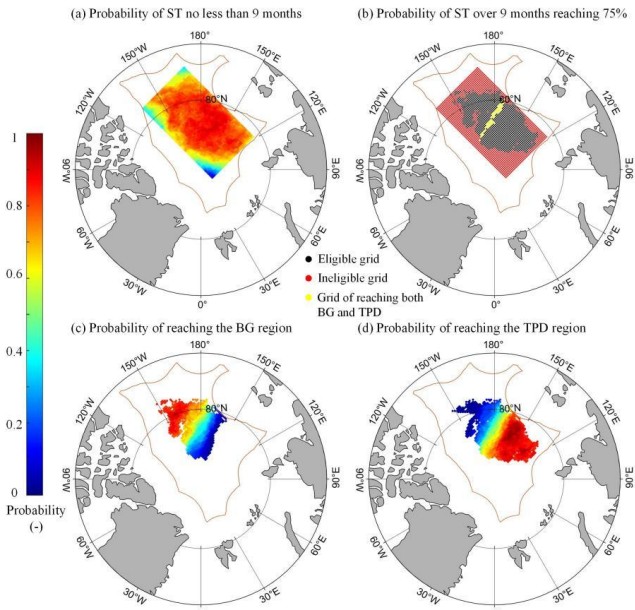


**Figure 4.** Spatial distribution of probability that sea ice drifting from a defined grid point satisfies the following conditions in 1979–2020:
(a) the ST within the central Arctic Ocean for no less than 9 months; (b) the region with the probability of ST over 9 months reaching 75%
(black dot), also shown is the junction zone between the BG and TPD regions (yellow strip), defined using the climatological SIM field;
and the probabilities with the destinations of reconstructed trajectories reaching the (c) BG or (d) TPD region.

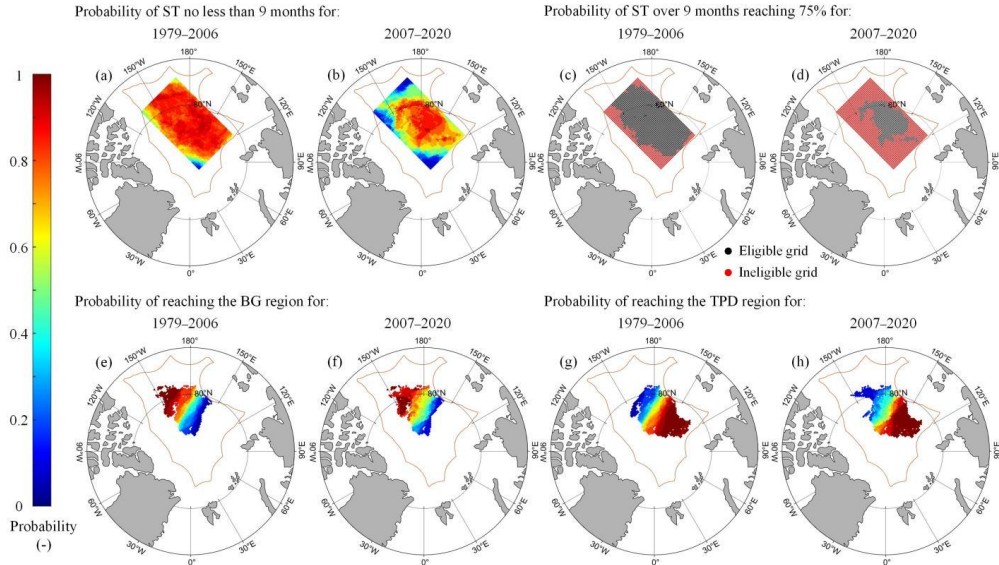

**Figure 5.** Spatial distribution of probability that sea ice drifting from a defined grid point in two sub periods of 1979–2006 and 2007–2020 under the following conditions: (a–b) probability of ST of not less than 9 months; (c–d) the region with the probability of ST over 9 months reaching 75% (black dot); and probability of reaching the (e–f) BG or (g–h) TPD region.

### 3.2 Changes in atmospheric thermodynamic forcing along the trajectories

Sea ice thermodynamic growth is regulated by both atmospheric and ocean forcing. Since the oceanic heat flux underneath the ice is relatively weak during the freezing season (Lei et al., 2022), near-surface air temperature could be considered as the most decisive parameter regulating ice growth and is a major atmospheric forcing factor for the sea ice growth analysis model (Leppäranta, 1993). The Arctic Amplification can reduce ice growth during the freezing season (Ricker et al., 2017a) and trigger an earlier onset of sea ice melting (Stroeve et al., 2014). Although the reconstructed ice trajectories in the BG region are able to approach areas further south and experienced higher air temperatures, the air temperature along ice trajectories in the TPD region had a slightly higher increasing trend (0.081 °C/yr) than that in the BG region(0.078 °C/yr) in 1979–2020. This is consistent with the results given by Rantanen et al., (2022), which revealed a relatively high warming trend in the Atlantic sector compared to other regions in the Arctic Ocean, mainly due to the enhanced atmosphere-ocean heat flux caused by the reduction of the ice cover, enhanced warm air mass intrusions and changes in atmospheric circulation. Despite significant increases in air temperatures in both regions, the occurrence of extremely high air temperatures exceeding 90th percentile of daily mean from 1979 to 2020, also defined as hot days by Vautard et al., (2013), did not change significantly. These hot days occurred mainly in June, with negligible regional variation in frequency between the BG (7.3%–12.8%) and TPD (6.3%–13.2%) regions. This implies that extremely events



with high near-surface air temperature are largely concentrated in the initial stage of ice melting (Markus et al., 2009). These
events are often accompanied by the process of rainfall (e.g., Robinson et al., 2021), accelerating the melting of snow and
sea ice surfaces, promoting the formation of melt ponds (e.g., Feng et al., 2021), and triggering positive albedo feedback (e.g.
Goosse et al., 2018).

To further investigate the potential impact of changes in the warm-cold season transition along the trajectory on sea ice

melting or freezing, we also calculated the 30-day running average air temperatures and identified dates when the air
temperature rises above 0℃ and falls below –1.8℃ in 1979–2020. A significant delay trend ($P<0.05$) in the dates when
near-surface air temperatures fell below –1.8℃ was only observed in the BG region, indicating a delayed onset of ice
freezing, and possibly leading to increased multi-year ice melt in the summer there (Babb et al., 2023). No significant trend
has been identified in the seasonal transition from cold to warm for both region. As shown in Fig. 6a, the FDD of the ice
trajectories reaching the BG region was generally higher compared to those reaching the TPD from 1979 to 2020, which
suggests warmer conditions during the freezing season in the TPD region, although the ice trajectories in this region were
located in a relatively high-latitude area. Furthermore, the significant decreasing trend ($P<0.05$) in the FDD along the
trajectories is slightly greater relative to that in the BG, consistent with the larger warming trend in the TPD region. However,
the magnitude of TDD along the trajectories reaching both regions of BG and TPD did not differ considerably and did not
reveal a clear trend due to the unclear warming trend for the summer in the Arctic Ocean. In winter, the average surface net
longwave radiative flux along the trajectory in the TPD region was upward, indicating the heat loss from the sea ice-ocean
system to the low atmosphere, with the peak of probability distribution increasing from about –57.5 W/m$^2$ prior to 2007 to
about –50.0 W/m$^2$ after 2007. However, such shift in the BG region was relatively weak from about –57.5 W/m$^2$ to about
–52.5 W/m$^2$ (Fig. 7). This indicates that the weakened radiational cooling effect from the surface in the TPD region under the
clear-sky conditions was more pronounced compared to that in the BG region, which also can be attributed the difference in
the winter warming trend between two regions. Moreover, the frequency with the net longwave radiation feature under the
opaquely cloudy state during the winter, having the typical value of > –10 W/m$^2$ (Graham et al., 2017), increased from 3.5%
in 1979–2006 to 4.5% in 2007–2020 in the TPD region, while it decreased from 4.6% to 4.2% in the BG region. This can, at
least in part, be attributed to the more prominent trend of enhancement for the active cyclonic activity in the Atlantic sector
of Arctic Ocean than the western Arctic Ocean (e.g., Zhang et al., 2023).



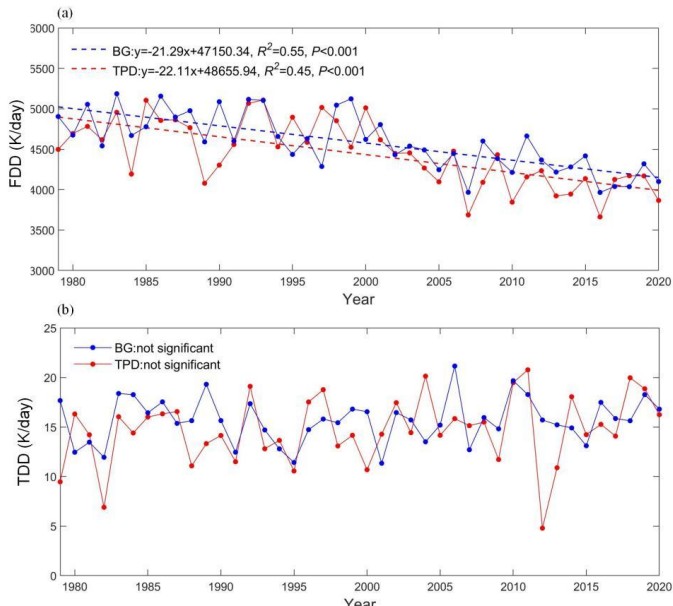


**Figure 6.** Changes in (a) FDD and (b) TDD during October to June in two regions between 1979 and 2020.

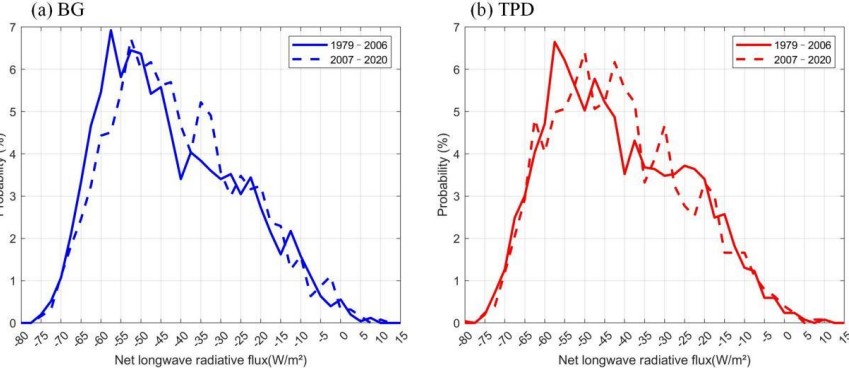


**Figure 7.** Probability distribution of winter surface net longwave radiative flux along the trajectories in the BG and TPD regions for two
sub periods of 1979–2006 and 2007–2020, with negative values denoting the upward flux from the surface.

**3.3 Changes in sea ice conditions along the trajectories**

The dynamic response of sea ice to wind forcing can be characterized using the ice-wind speed ratio (Herman and

Glowacki, 2012). As shown in Fig. 8, the seasonal average ice-wind speed ratio along the ice trajectories was largest in
autumn, which may be due to the relatively weak sea ice consolidation at that time (e.g., Lund-Hansen et al., 2020). In this
season, the ice-wind speed ratio was slightly larger in the TPD region (1.53%) than in the BG region (1.48%), which is
consistent with the TPD region being generally considered as a region with a higher ice-wind speed ratio than other regions



in the Arctic Ocean (Haller et al., 2014). Note that, the ice-wind speed ratio obtained from this study were slightly lower than
those obtained from buoy observations close to the North Pole by Haller et al., (2014), as the remote sensing SIM product
typically underestimates SIM speeds due to the low temporal resolution (Gui et al., 2020). The increasing rate of the
ice-wind speed ratio along the trajectory in the BG region is larger than that in the TPD region in all seasons. This suggests
that sea ice drift in the BG region is undergoing a period of progressively stronger response to wind speed. This is because
the thick multi-year ice with weak mobility there has been gradually replaced by the thin seasonal ice (Babb et al., 2023).
Thereby, Lagrangian observations in the BG region would experience an ongoing enhancement of dynamic response of sea
ice to wind forcing compared to the TPD region. Especially since 2007, as the acceleration of SIM has been more apparent
(Sumata et al., 2023), the seasonal average ice-wind speed ratio in the BG region increased to 1.54±0.2% for autumn, winter
and spring seasons, which already overwhelmed those of the TPD region by about 10%. Therefore, from the perspective of
sea ice dynamics response, the observation data of the SHEBA campaign (e.g., Lindsay, 2002) are not representative of the
current ice state in the BG region.

Additionally, we estimated the SIT along the ice trajectories to evaluate trends and spatial differences in ice conditions.

During the period from October to April in 2010–2020, with the CryoSat-2/SMOS SIT data available, there is no significant
trend in the SIT along the trajectories for both regions of BG and TPD (Fig. 8b). In most years (64%), especially during the
early stage (autumn) of ice season, the ice along the trajectories in the TPD region was thinner than that in the BG region.
This is likely because the younger ice age for the ice floes at the deployment areas drifting into the TPD region finally, which
is highly possible originating from the polynyas in the Laptev Sea or the East Siberian Sea (e.g., Krumpen et al., 2020).
Furthermore, the SIT standard deviation along trajectories reaching the TPD region was higher than that in the BG region in
most years (73%), implying that the spatial variation in SIT along trajectories reaching the TPD region is greater.

It is noteworthy that in 2014 and 2018, the SIT anomalies in the BG and TPD regions were opposite, with the positive

(negative) values in the BG (TPD) region compared to the 2010–2020 mean. This may be related to the atmospheric
circulation anomalies. In 2014 and 2018, the BH index was slightly higher than the 1979–2020 average. Accordingly, the
above-average SLP over the BG region strengthened the anticyclonic circulation of sea ice and favored more ice to be
trapped there. As a result, less sea ice can be advected from the western Arctic Ocean to the TPD region. Consequently, the
ice-wind speed ratios along the ice trajectories in the BG region in these two years decreased to the first and second smallest
in 2010–2020, which suggests that the increase in SIT in the BG region gives a precondition for the reduced sea ice response
to wind forcing. However, such connection does not occur in the TPD region. This is likely because the relatively thin ice
was only observed in the early freezing season in the TPD region, with the average SIT in October 2014 (0.80 m) and 2018
(1.36 m) was 54.0% and 92.3% of the 2010–2020 average SIT. This difference would be quickly alleviated by the recovery





of SIT due to the thin ice-growth rate negative feedback (e.g., Lei et al., 2022).

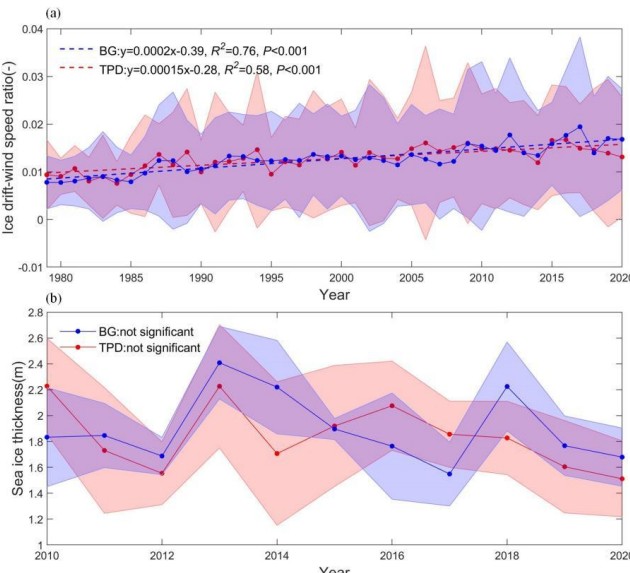


**Figure 8.** Changes in (a) ice-wind speed ratio in 1979–2020 and (b) ice thickness in 2010–2020 along trajectories reaching the BG or TPD region.

## 4. Discussion

### 4.1 Assessment of reconstructed sea ice drift trajectories

To examine the reliability of the method for the ice trajectory reconstruction using the remote sensing SIM product, we
used data from 10 buoys in each of the BG and TPD regions as validation data, respectively. The selected buoys were mostly
deployed from September to November, with a measurement duration ranging from 2 to 12 months. The deployment time is
roughly consistent with the start time of our reconstructed trajectories. For comparison, ice drift trajectories were
reconstructed from the initial deployment locations of the buoys.
The results reveal that within the initial 100 days, most misalignment distances were less than 50 km, as shown in Fig.
9a). However, the reconstructed ice trajectories form the deployment sites of 5 buoys were misaligned with the buoys
trajectories by a relatively large distance, with the values of about 146.3–173.0 km after 100 days. This may be related to the
complex wind conditions at the early stage of drifting of these buoys. They experienced relatively high wind speeds for most
of the initial 10 days of the drift, with an abnormal wind speed of 6.8±2.7 m/s related to the climatology since 1979 (5.6±0.4
m/s). Conversely, our reconstructed ice trajectories reaching the regions of BG and TPD have initial 10-day averaged wind



speeds of 5.6±2.5 and 5.8±2.8 m/s over the 43-year period, respectively. This indicates that most reconstructed trajectories
in this study did not experience such strong complex wind conditions that affect the reconstruction accuracy. In addition, the
SIC at the initial location of the reconstructed ice trajectories also can affect the reconstruction accuracy to some extent, with
relatively large misalignment distances for the low SIC. Fortunately, all the reconstructed ice trajectories, as shown in Fig. 3,
have starting points with SIC above 95% over the 42-year period, which could greatly reduce the influence of SIC on the
reconstruction accuracy in this study.
Excluding these 5 buoys mentioned above with complex initial wind conditions, the average misalignment distances
between the reconstructed trajectories and the buoy trajectories (9 cases) after 9 months are 60.2±40.8 km (about 2.5 pixels
of ice motion product), with cosine similarities all above 0.94, and the mean deviation of the Euclidean distances of
endpoints is 119.8±85.9 km. This implies that the geometric similarity between the reconstructed trajectories and the buoy
trajectories is highly consistent, which can ensure the appropriate direction and destination of the reconstructed ice trajectory.
Regionally, there is a better reconstructed trajectory performance in the BG region than that in the TPD region. After 9
months, the average misalignment distance in the BG region (2 cases) is about 17.1 km, which is 23.6% of that in the TPD
region (7 cases), consistent with the visual comparison of drift trajectories shown in Fig. 9b. This may be due to the larger
SIM speed and its meridional gradient in the TPD region, especially in the southern region. All reconstructed trajectories in
the TPD region terminate at a further northward location compared to the corresponding buoy trajectories. This leads to a
relatively adventurous estimate of the effective duration of ice trajectories in the TPD region. However, we estimate this
uncertainty to be approximately 2.1% (or 5.7 days) compared with the buoy measurement, which looks trivial.

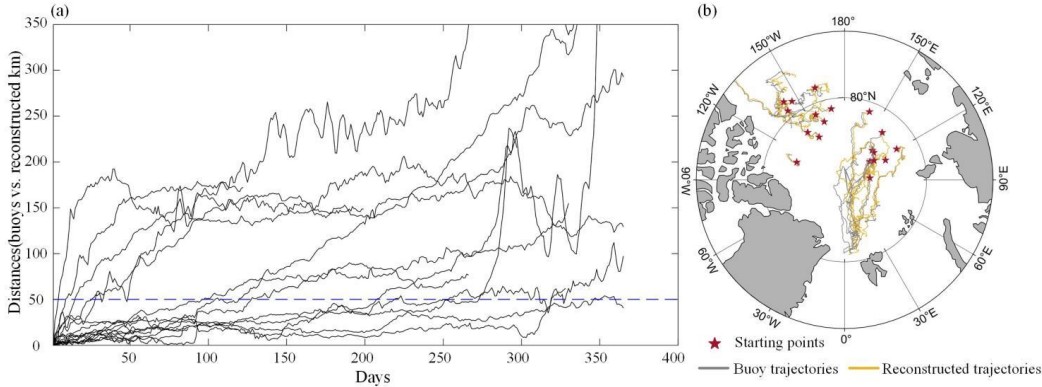


**Figure 9.** Comparison of reconstructed ice trajectories with buoy trajectories: (a) misalignment distances over time for trajectory pairs; (b)
trajectory pairs from the deployment site of buoys.



### 4.2 Links of endpoints of ice trajectories to atmospheric circulation patterns

Based on three scenarios for ice trajectory endpoints, i.e., > 90% probability of reaching the BG or TPD region and 40–60% probability of reaching both regions, we explored the links of endpoints of ice trajectories to atmospheric circulation patterns (Tables 1–2). We also analyzed the statistical relationship between the distance from the endpoint to Fram Strait (80° N) for the ice trajectories reaching the TPD region and atmospheric circulation indices, which allows exploring the potential impact mechanisms of atmospheric circulation patterns on the sea ice outflow from the central Arctic Ocean.

For ice trajectories reaching the TPD region, AO can significantly explain 11.4% of the endpoint longitude after 6 months using the winter index and performed insignificantly on an autumn scale, which is probably because AO is generally strong in winter (Rigor et al., 2002). Conversely, autumn CAI, DA, and BH, can explain 16.2%–31.1% ($P<0.05$) of the endpoint latitude after 9 months. These indices were also significantly correlated with the distance between the endpoint of the ice trajectory after 9 months and the Fram Strait ($R^2$: 10.2%–36.0%, $P<0.05$). When the BH index is positive, the BG squeezes the axis alignment of the TPD eastward, lengthening the distance that ice advects along the TPD toward the Fram Strait. Compared to BH, the autumn CAI and DA were more strongly correlated with the endpoint latitude or the distance between the endpoint and the Fram Strait after 9 months. As their positive phases imply enhanced meridional wind forcing in the TPD region, which exacerbates the transpolar sea ice drift (Wu et al., 2006; Vihma et al., 2012). Therefore, it is necessary to take the autumn CAI and DA index into consideration for predicting the subsequent trajectory of ice floe, as well as the buoy or camp deployed on it in the TPD region.

For the ice trajectory reaching the BG region, all atmospheric circulation indices in autumn had a significant impact on the endpoints after 9 months, with endpoint longitude being significantly influenced by AO, CAI, DA, and BH ($R^2$: 10.0%–25.0%, $P<0.05$) and endpoint latitude being significantly correlated with CAI, DA, and BH ($R^2$: 11.2%–25.6%, $P<0.05$). Although sea ice circulation in the BG region is driven by anticyclonic wind stress curl associated with the positive BH index (Proshutinsky et al., 2002), the BH did not reveal more effective interpretability for the location of the ice drift endpoints in this region than other indices. Moreover, for sea ice that has the potential to reach both regions of BG and TPD, all atmospheric circulation indices in autumn also had a significant explanatory level for the latitude of endpoint and its distance from the Fram Strait after 9 months ($P<0.05$).

Since atmospheric circulation patterns during the start stage of ice drift in autumn, especially for the CAI and DA, had a strong influence on the endpoints of ice trajectories after 9 months in both regions of BG and TPD, we further analyzed scenarios where these indices exhibit extreme positive (negative) anomalies, defined with the value higher (lower) than the 1979–2020 climatology by one standard deviation (Fig. 10). When CAI and DA are at extreme positive (negative) phases,





the spatial proportions of starting points with a 9-month ST threshold for more than 75% years are 75.2% (46.3%) and 86.0%
(44.1%), respectively, which are greater (less) than the spatial proportions obtained from the mean field in 1979–2020
(53.2% as shown in Fig. 2). This suggests that the extreme scenarios of autumn CAI and DA have a pronounced modulating
effect on the ideal deployment areas for Lagrangian observations, with a wider range of ideal areas at their extreme positive
phases. Under extremely positive phases of CAI and DA, the preferred area of deployment tends to extend to the Chukchi
Sea and the Canada Basin, while at the negative phase it prefers the northern Laptev Sea. However, the extremely positive
phase of the autumn CAI only favors a trivial increase (by 0.5%) in the spatial proportion of points with > 75% probability
of reaching the TPD region compared to the average state over 42 years. The extreme negative phase of the autumn DA, on
the other hand, significantly increases the probability of reaching the BG region, and the spatial proportion > 75% is 1.4
times that obtained from the whole study period.
**Table 1.** Coefficient of determination ($R^2$) between atmospheric circulation indices and location (longitude/latitude) of sea ice trajectory
endpoint after 9 months in 1979–2020.

| | Regional tendency | Autumn CAI | Autumn AO | Autumn DA | Autumn BH | Winter AO | Winter CAI | Winter DA |
|---|---|---|---|---|---|---|---|---|
| Longitude of endpoint | TPD | n.s. | n.s. | n.s. | n.s. | 0.114 | 0.093 | 0.096 |
| | BG | **0.250** | *0.237* | 0.100 | 0.146 | n.s. | n.s. | n.s. |
| | TPD/BG | n.s. | n.s. | n.s. | n.s. | n.s. | 0.103 | n.s. |
| Latitude of endpoint | TPD | **0.286** | n.s. | **0.311** | *0.162* | n.s. | n.s. | n.s. |
| | BG | *0.166* | n.s. | **0.256** | 0.112 | n.s. | n.s. | n.s. |
| | TPD/BG | **0.242** | 0.115 | **0.266** | 0.111 | n.s. | n.s. | n.s. |

Note: Significance levels are $P < 0.001$ (bold), $P < 0.01$ (italic) and $P < 0.05$ (plain); n.s. denotes insignificant at the 0.05 level.
**Table 2.** Coefficient of determination ($R^2$) of atmospheric circulation indices for the distance from the sea ice trajectory endpoint after 9
months to the Fram Strait in 1979–2020.

| Regional tendency | Autumn CAI | Autumn AO | Autumn DA | Autumn BH |
|---|---|---|---|---|
| TPD | **0.359** | n.s. | **0.360** | 0.102 |
| TPD/BG | **0.295** | 0.137 | **0.299** | 0.123 |

Note: Consistent with Table 1.

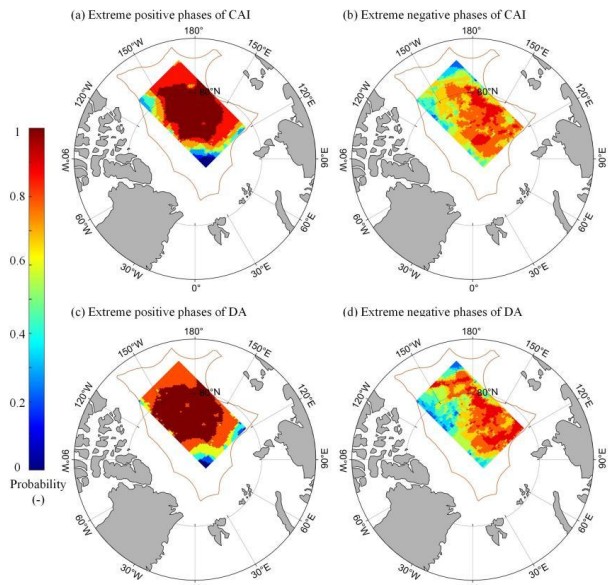


**Figure 10.** Spatial distribution of the probability that the ST of sea ice drifting from a defined grid point is not less than 9 months at the extreme positive and negative phases of the autumn CAI and DA for 1979–2020.

### 4.3 The ST of ice trajectories disregarding the EEZ boundary

The ST of reconstructed ice drift trajectories, or the potential Lagrangian observations on the basis of ice camp and buoy is limited to a high extent by the EEZ boundary. To quantitatively evaluate the impact in this regard, we hypothesized a desirability of enhanced international cooperation to reduce the impact of geopolitical boundaries on this type of observations, and identified the ideal deployment areas for Lagrangian observations under this scenario. It is found that the probability of ST exceeding 9 months for ice trajectories reconstructed from all grids in the rectangular study region ranged from 47.6% to 92.9% between 1979 and 2020 without the limitation of the EEZ boundary. The spatial range with probabilities > 75% (i.e., 32 years) for the ST threshold of 9 months extends to 89.6% of the whole rectangular study region, much larger compared to that (53.2%) with the limitation of the EEZ boundary. Disregarding the EEZ boundary, the increase in eligible starting points with > 75% probability is proportional to the used ST threshold (Table 3). Especially for the 10-month ST threshold, the eligible area increases by over 200% compared to that with the limitation of the EEZ boundary. Disregarding the EEZ boundary, the increase in eligible starting points in the rectangular study region with > 75% probability of reaching the BG or TPD regions is also proportional to the ST threshold. Particularly for the 10-month ST, the number of eligible starting points reaching the BG or TPD region increases by over 100% through removing the limitation of EEZ boundary. For starting points with a close probability of reaching both regions of BG and TPD, the spatial proportion of





eligible starting points would instead be suppressed compared to that estimated with the consideration of the EEZ boundary.
This is because these eligible starting points are primarily located at the junction of two regions of BG and TPD and
relatively far from the EEZ boundary.
For the period 1979–2020, the average Lagrangian observation duration in the rectangular study region disregarding the
EEZ boundary is about 335.7±77.7 days, which extends by about 8.9 days compared to those estimated with the
consideration of the EEZ boundary. Regionally, the Lagrangian observations located in the BG and TPD regions would be
further extended by about 10.6 days (336.1±77.4 days) and 7.0 days (335.2±77.9 days), respectively. This suggests that the
EEZ boundary has a slightly larger impact on the observation duration in the BG region compared to the TPD region,
because the EEZ boundary in the downstream of TPD is overall close to the marginal ice zone. Spatially, for sea ice reaching
the BG region, the added eligible starting points are located in the southern part of the BG, as shown in Fig. 11. Sea ice
originating from these locations might be more strongly affected by the clockwise ice circulation of the BG and cross beyond
the EEZ boundary in the south more easily. For the ice trajectories reaching the TPD region, the added eligible starting
points are located in the south of the study region or in the sector facing the Fram Strait. Sea ice originating from these areas
might have been advected more rapidly to cross the EEZ boundary in the Atlantic sector.
**Table 3.** Increased spatial percentage in eligible starting points without considering the EEZ constraints compared to those estimated with
the constraints.

| ST threshold in the ice zone (months) | 6 | 7 | 8 | 9 | 10 |
|---|---|---|---|---|---|
| Case 1: probability of ST over corresponding ST > 75% | 26.4% | 36.2% | 50.2% | 68.4% | 208.1% |
| Case 2: probability of reaching the BG region > 75% | 29.5% | 40.0% | 63.5% | 103.4% | 195.1% |
| Case 3: probability of reaching the TPD region > 75% | 12.1% | 25.5% | 36.6% | 51.4% | 198.5% |
| Case 4: probability of reaching the BG or TPD region ranging between 40% and 60% | –25.9% | –35.6% | –31.1% | –37.1% | –50.0% |

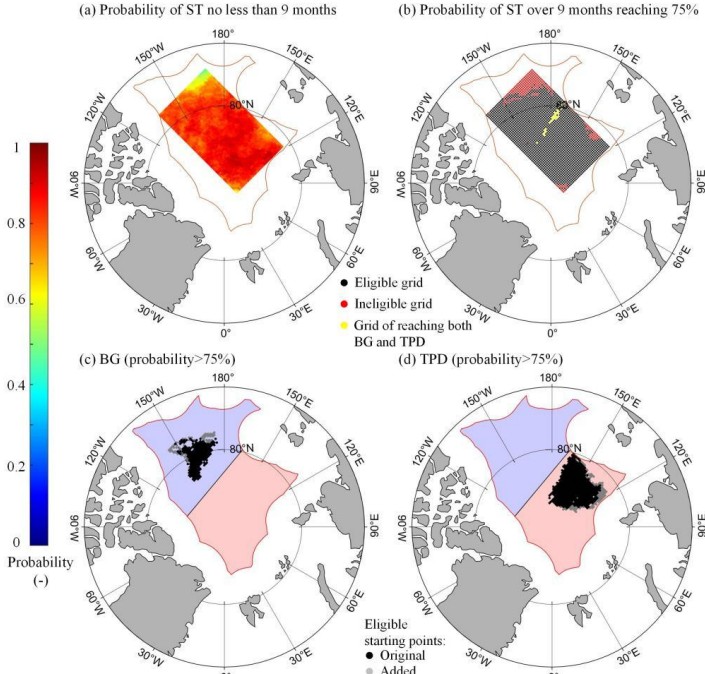

**Figure 11.** Assuming that the EEZ boundary constraints are not considered in 1979–2020: (a) spatial distribution of the probability of ST in the ice region not less than 9 months; (b) the region with the probability of ST over 9 months reaching 75% (black dot), also shown is the junction zone between the BG and TPD regions (yellow strip); and (c-d) the added eligible starting point (gray) with > 75% probability of reaching the BG or TPD region, compared to those estimated with constraints (black).

**4.4 The influence of deployment time**

In this study, we estimated the potential Lagrangian observation duration relying on Arctic ice floes, based on the deployment commenced on October 1 each year. On the one hand, it is based on our general understanding of the thaw-freezing annual cycle of Arctic sea ice. That is to say, the sea ice in the central Arctic Ocean enters a new growth period from the end of September or early October onwards every year. On the other hand, this is based on the experience of the MOSAiC. Here, we further test the influence of deployment time on the estimated duration of Lagrangian observation.

Using the starting points reaching the BG or TPD region over 90%, as shown in Fig. 3, we further calculated the duration with the deployment date ranging from August 15 to November 1 to explore the influence of setup time on the potential duration of subsequent observations (Fig. 12). The mean duration of Lagrangian observation in the two regions decreases gradually from 301.6±17.2 days for the deployment on August 15 to 282.3±22.9 days according to the deployment on November 1. Although the advanced deployment of ice stations or buoys based on ice floes to August 15 may result in



longer observation time, approximately by 11.8 days, compared that derived from the deployment on October 1. We still
argue that it is more appropriate to implement the deployments of ice camps or buoys over the Arctic ice floes in October if
the logistics support allows, because there is often a risk that the ice holes drilled for the equipment deployment in August
and September are hard to refreeze, and the risk of floe fragmentation will increase at the end of ice melt season. In these
situations, the equipment is prone to collapse, causing observation interruptions. As expected, the duration in both regions is
longer in the case of disregarding the EEZs relative to that derived with the EEZ restriction. Even in the BG region, with a
shorter duration of observation compared to that in the TPD region, the potential duration for Lagrangian observation is
estimated to reach 253.1 days with the EEZs and 276.1 days without the EEZs, respectively, with the deployment on
November 1. This suggests that the deployment of buoys or camps on the floes in the central Arctic Ocean, even by the end
of October, is still able to guarantee a observation duration of at least 8 months.

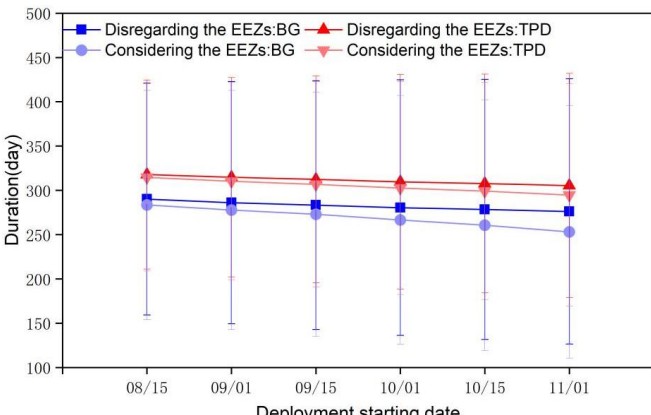


**Figure 12.** Changes in the mean duration of Lagrangian observations in 1979–2020 for various deployment dates from the starting points
reaching the BG or TPD region over 90% , for both cases of taking into account or disregarding the EEZs.
**5. Conclusions**
From a rectangular study region defined in the central Arctic Ocean excluding the EEZs, we reconstructed the sea ice
trajectories from 1979 to 2020 and determined the ideal deployment areas for the subsequent Lagrangian observations with a
an expected duration. On this basis, regional differences in the atmospheric conditions and response of ice dynamics to wind
forcing along the trajectories were assessed. Subsequently, we explored the regulation mechanisms of atmospheric
circulation patterns on sea ice advection and the influence of EEZ boundary constraints and deployment time on the duration
of sustained Lagrangian observation.
Deployment of Lagrangian observations at locations centered around 82°N and 160°E, near the north of East Siberian





and Laptev seas, can ensure at least 9 months of drifting observation time, with probabilities of remaining in the central
Arctic ice region ranging from 76.2% to 92.9% during the 42-year study period. Ice floes originating from this area of
$7.6 \times 10^5$ km$^2$ are more likely to reach the TPD region.

There are obvious regional differences in the atmospheric and sea ice conditions during ice drifting between the BG and

TPD regions. Near-surface (2 m) air temperatures in both regions of BG and TPD show a significant warming trend in
1979–2020, with a higher increasing rate in the TPD region than in the BG region due to its proximity to the Atlantic sector
of the Arctic Ocean. The significant decrease in FDD in the BG and TPD regions suggests that sea ice has experienced
warmer conditions during the freezing season in recent years. Lagrangian observations in the TPD region would experience
increased days of cloud opacity during the winter 2007–2020 by 28.5% compared to that in 1979–2006, because the cyclone
activities are more frequent in the TPD region in recent years. From a dynamic perspective, the observations in the TPD
region in early years would experience a relatively strong dynamic response of sea ice to wind forcing, with a higher
ice-wind speed ratio than in the BG region. However, this response has been enhanced more prominently in the BG region
due to the larger loss of sea ice, especially for the south part of BG region. Large-scale atmospheric circulation patterns at the
early stage of ice drifting in autumn have a significant influence on the terminal location of ice trajectories. Thus, compared
to the 1979–2020 average, the extreme positive phases of CAI and DA indices in autumn would expand the ideal deployment
area to the Chukchi Sea and the Canada Basin. On the contrary, at the extreme negative phase of these indices, it is preferred
to expand to the northern Laptev Sea.

In addition to natural conditions, the EEZ boundary has a great constraint on the Lagrangian observations. The absence

of these constraints would increase the number of eligible starting points in the study region. Disregarding the EEZ boundary
constraints, the eligible starting points with the trajectories toward the BG region expands southward, while for those toward
the TPD region it would expand in the areas facing to the Fram Strait. The advanced deployment start time to mid August
may result in a longer duration of Lagrangian observations, by 11.8 days compared to that obtained from the deployment on
October 1. However, in order to reduce the failure risk of observation instruments deployed on the floes, in particular in the
later ice melt season, we still consider the deployments in October are more appropriate for Lagrangian observation relying
on ice floes in the central Arctic Ocean. The accuracy of reconstructed ice trajectories might be affected by low SIC,
complex windy weather at the initial location. However, we argue the influence of SIC and wind conditions on the
reconstructed ice trajectories used in this study is relatively unremarkable, because the initial stage of our reconstructed
trajectories has relatively high ice concentrations and relatively low wind speeds, both of which are beneficial for reducing
the uncertainty of ice-trajectory reconstruction.



In this study, daily SIM product is the main data source used to reconstruct sea ice drift trajectories and evaluate the ST
of Lagrangian observations relying on ice floe. We acknowledge this as a primary evaluation, ignoring operational safety
risks. The main challenges for survival and maintaining continuous observation for the specific devices deployed on the
Arctic ice floes include the breakage or compression of sea ice, the formation of melt ponds, and the intrusion of polar bear,
etc. As Arctic warming continues, the combined effects of accelerated melting and limited replenishment of multi-year ice
will eventually trigger the complete loss of multiyear ice and a shift to a seasonally ice-free Arctic ocean (Babb et al., 2023).
This change puts forward greater demands on ice floe-based observational campaigns and on the development of more
adaptive observational techniques and equipment to cope with future extreme ice and atmospheric environments. Our work
mainly provides supporting information for the site selection for the deployments of ice buoy and ice camp. The preferred
areas identified in this study still require adaptable adjustments, associated with the changes in Arctic sea ice itself in the
future. From a practical perspective, once reaching the preferred deployment area, the specific conditions of the ice floe,
such as ice thickness, floe size, distribution of ice ridge and melt pond, need to be further surveyed using high resolution
satellite remote sensing images and helicopters or ice-based measurements.
**Appendix**
**Table A1.** Basic information on buoy data used for validation of reconstructed ice drift trajectories

| Number | Start date (YY/MM/DD) | Start location (°N, °E) | End Date (YY/MM/DD) | End location (°N, °E) | Duration (Day) | Buoys type |
|--------|-----------|----------------|----------|--------------|----------|------------|
| 1 | 18/10/01 | 78.49, -146.12 | 19/08/24 | 71.29, -133.35 | 328 | Snow_Buoy |
| 2 | 20/11/04 | 83.93, -149.12 | 20/12/30 | 82.53, -144.07 | 57 | iSVP |
| 3 | 20/11/04 | 83.77, -110.26 | 20/12/30 | 82.81, -115.35 | 57 | iSVP |
| 4 | 20/11/04 | 82.50, -160.67 | 20/12/30 | 81.18, -154.30 | 57 | iSVP |
| 5 | 20/10/01 | 79.12, -140.50 | 20/12/26 | 76.66, -141.98 | 87 | ITP |
| 6 | 18/08/13 | 81.19, -169.34 | 19/02/27 | 80.88, -134.24 | 199 | SIMBA |
| 7 | 14/09/01 | 77.96, -141.98 | 15/05/24 | 75.67, -151.84 | 266 | iSVP |
| 8 | 14/09/01 | 81.32, -156.03 | 15/08/31 | 77.85, -138.64 | 365 | iSVP |



| 9 | 14/09/01 | 78.24, -162.07 | 15/08/31 | 79.50, -151.95 | 365 | iSVP |
|---|----------|----------------|----------|----------------|-----|------|
| 10 | 16/09/01 | 82.67, -142.03 | 16/12/31 | 77.99, -132.51 | 122 | iSVP |
| 11 | 15/10/01 | 85.06, 136.82 | 16/09/30 | 83.28, 8.21 | 366 | PAWS |
| 12 | 15/10/01 | 84.46, 115.64 | 16/09/12 | 81.13, 5.95 | 330 | iSVP |
| 13 | 15/10/01 | 85.06, 136.92 | 16/09/30 | 83.27, 8.20 | 366 | Snow_Buoy |
| 14 | 18/10/01 | 82.63, 141.50 | 19/08/26 | 82.42, 11.34 | 330 | iSVP |
| 15 | 18/10/01 | 81.17, 159.90 | 19/08/24 | 87.18, 13.64 | 328 | Snow_Buoy |
| 16 | 19/10/01 | 82.62, 120.56 | 20/09/29 | 83.30, 8.73 | 364 | iSVP |
| 17 | 19/10/01 | 86.18, 125.61 | 20/06/08 | 81.05, 3.78 | 252 | iSVP |
| 18 | 19/10/10 | 85.13, 133.02 | 20/07/14 | 81.04, -0.10 | 279 | SIMBA |
| 19 | 19/10/01 | 85.71, 123.25 | 20/07/14 | 81.06, -0.67 | 288 | SVP5S 003 |
| 20 | 19/03/26 | 86.90, 94.19 | 19/12/08 | 81.11, 4.56 | 258 | iSVP |

**Data Availability**
Sea ice motion, concentration data from NSIDC is available at https://nsidc.org/data/NSIDC-0116/versions/4.and
https://nsidc.org/data/G02202/versions/4.    Sea    ice    thickness    data    is    downloaded    from
https://data.seaiceportal.de/data/cs2smos_awi/v204/. Shapefiles of maritime boundaries and EEZs are publicly available
online    (https://www.marineregions.org/).    The    ERA5    reanalysis    data    are    downloaded    from
https://cds.climate.copernicus.eu/cdsapp#!/dataset/reanalysis-era5-single-levels.    Buoy    data    is    available    at
https://www.meereisportal.de/.
**Financial support**
This work was financially supported by the National Natural Science Foundation of China (grant No. 42325604), the
Ministry of Industry and Information Technology of China (grant No. CBG2N21-2-1), and the Program of Shanghai
Academic/Technology Research Leader (Grant No. 22XD1403600).



**Competing interests**

The contact author has declared that none of the authors has any competing interests.

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
