# Peer review of "Estimation of duration and its changes in Lagrangian observations relying on ice floes in"

_EGUsphere, 2024_

## Referee Comment (RC3)

Review of "Estimation of duration and its changes in Lagrangian observations relying on ice floes in the Arctic Ocean utilizing sea ice motion product"

The paper focuses on improving Arctic Lagrangian observations by analyzing long-term sea ice motion data (1979-2020). The study evaluates suitable deployment zones for ice camps and buoys by using sea ice motion products and incorporating atmospheric circulation patterns like the Arctic Oscillation and Arctic Dipole. The authors highlight the declining survival time of ice floes and the increasing challenges for Lagrangian observations due to climate change. This research is highly relevant, given the rapid transformations in Arctic sea ice dynamics and the growing need for precise observational data for climate modeling. By integrating trajectory simulations with EEZ constraints, the study provides actionable insights for future observational campaigns.

I appreciate the exhaustiveness of all sea ice thermodynamic and dynamic throughout the manuscripts. However, I have several concerns regarding the Area Of Insterest (AOI) , the methodology protocol, and some logic explanations. Therefore, I recommend that the paper undergo major revisions before it can be considered for publication.

**General Comments**:

1. I realized that the aim of the work is to provide the reference for the ideal deployment locations in the central Arctic Ocean (in Line 73), but I don't understand why author choose the starting points region just within the rectangular area instead of within the EEZ boundary since EEZ anyway is devided into BG and IPD? So, I am not sure the motivation, is it just want to find the ideal depolyment region within the rectangular only?

2. Data and method part: How do you interpolate the 25 km ice motion when employing Lagrangian methods, linear or inverse distance weighting? Do you apply the Lagrangian method from start to end without any regridding during the period? How do the results compare to a semi-Lagrangian approach?

3. The validation of buoy trajectories seems to focus on data after 2014. Are there additional buoy datasets available from earlier periods? If not, are the selected buoys representative and exhaustive for this study?

4. Another interesting point to explore could be backtracking trajectories instead of forward tracking. For trajectories with >9 months survival time (ST), does the backtrack reveal that their starting points are mostly within the rectangular AOI? This may provide valuable insights into uncertainties and trajectory origins.

5. When using 2m air temperature for calculating Freezing Degree Days (FDD), how was the daily value derived - was it simply a mean of hourly data? Providing clarity on this calculation is crucial for reproducibility. How about the bias in ERA5 temperature.

6. I am more interested in Figure 5, which is more pratically in the future. Shouldn't you further add more recommendation on the depolyment for the future based on the 2007-2020 analysis (and also, could you longegate the time span from 1979-2023), and further make some uncertainties or high-recommend and midiate-recommend about the region? Since now for me, the all materials somehow distract me about the whole movitation. Incorporating uncertainty estimates and differentiating regions into high-recommendation and moderate-recommendation zones would greatly enhance the practical utility of the paper. As it stands, the extensive materials somewhat distract from the core motivation of the study.

7. Section 3.3, I'm not sure how much information related to the motivation can ge obtained from here, please considering make them concrete.
8. Section 4.1 requires further elaboration. In particular, I recommend adding an uncertainty analysis or sensitivity test to strengthen the robustness of the findings.
9. I don't fully capture the Table 1 concerning its physical mechanism, first of all, how to understand the autumn CAI only have the obvious significant correlation with longitude in BH, but more correlated with both IPD and IPD/BH in latitude.

**Specific Comments:**

Line 23: change to "as the sea ice thins"

Line 117: use "optimal" instead of "most optimal"

Line 308-309: I am not sure about the statement since we don't know the casuality between ice motion, wind circualtion, near surface ocean current/stress. It is truly that sea ice motion, wind speed, ocean surface stress increase with climate change, but correlation doesn't give us some ideas in who is the trigger and who is the influencer. Could you provide more evidence.

Line 347: "form" to "from"

Line 393-395, can you explain why?

---

## Author Comment (AC1)

**Response to RC1**

Thank you for your time and constructive comments on the manuscript "Estimation of duration and its changes in Lagrangian observations relying on ice floes in the Arctic Ocean utilizing sea ice motion product". We would consider comments carefully and incorporate practically all of them in the revised manuscript.

**Comments:**

This manuscript is interesting to me, in which the ideal deployment locations in the central Arctic Ocean for ice camp or buoy have been discussed by using SIM product, and the main standard is to ensure that Lagrangian observations can last a period as long as possible. The logic and structure of the manuscript are ok for me, but I still suggest the authors should address the following issues.

(1) L66, "The Arctic sea ice is mainly driven by wind and oceanic current stresses". Maybe you mean the drift of Arctic sea ice or sea ice dynamics is driven by….

**Reply:** Yes, we will revise this sentence to "Arctic sea ice dynamics is mainly driven by wind and oceanic current stresses".

(2) L73-77 tell the motivation of this study. But I am not sure if there is any similar study on this topic? From the statement here, this study is the first one considering the ideal deployment location in Arctic Ocean.

**Reply:** We will consult more literature to confirm if similar studies have been done before. Meanwhile, in the introduction, we will also provide a broader overview and summary of the main achievements of previous studies in related work.

(3) The content in the introduction section is a little confusing. What exactly you want to summary in this part? Sea ice dynamics? Pervious ice stations? Or something else?

**Reply**: Thank you for pointing it out. Actually, what we want to emphasize is history and challenges of long-term observations based on ice camps or buoys,as well as the importance of identification of ideal deployment areas to ensure continuity and effectiveness of observations. We will further improve the relevant expressions.

(4) L110. The definition of the rectangular area in Fig.1 is still somewhat arbitrary to me. Maybe you can put the EASE-Grid as the background and then select some from all of them. "The reasons for this diagnosis will be given later." Please specify where you have discussed this problem.

**Reply:** Thanks to your suggestion. To specify this definition to avoid any arbitrariness, we will add the description of the selection of rectangular areas with the EASE-Grid grid as background. The detailed reasoning for this selection is discussed

in lines 211-214, and we will revise these sentences to make the explanation more explicit and accessible.

(5) Section 2.3. If the ice floe is broken into pieces during drifting, is there any impact on the calculation of the survival time (ST)?

**Reply:** In reality, the breakup of an ice floe during drift certainly has an effect on survival time. However, in our calculations, we cannot be judged when and where the ice floes break up using the sea ice motion product. In this study, we mainly use sea ice concentration (SIC) and exclusive economic zone (EEZ) boundaries to determine survival time. We have already explained such limitations of this study in lines 518-522, 525-530. Based on our analysis and diagnosis, we wish to identify potential ideal areas for buoy or ice camp deployments. In actual operation, the operating time of buoys or ice camps also depends on the breakup or collapse of the ice camp or buoy and its supporting ice floe, the formation of melt ponds and the intrusion of polar bear, etc. Therefore, what we infer should be the maximum potential survival time. However, we also argue that such analysis is still necessary for the selection of deployment areas for buoys or ice camps, especially when we hope to obtain longer observation time series.

(6) L197, "FDD(TDD) refers to the integral of near-surface air temperatures below…". What kind of air temperature? hourly average or daily average?

**Reply:** We use the daily average air temperature for the integration of FDD and TDD, which we will clarify in the manuscript.

(7) Section 3.2. Air forcing such as temperature and long-wave radiation have been investigated here. How about the precipitation? Snowfall poses an important impact on sea ice growth and decay.

**Reply:** Thanks for your suggestion, we will add some discussions of regional differences of the precipitation and snowfall and their influence on sea ice growth or decay.

(8) Section 3.3. The ice-wind speed ratio in Fig.8 is also overall lower than the typical values in free-drift analytical solution. You can also discuss this difference, maybe relating to sea ice concentration.

**Reply:** Thank you for the suggestion. We will further discuss why the ice-wind speed ratios are lower than typical values in free-drift analytical solutions, which are likely related to sea ice consolidation.

---

## Author Comment (AC2)

**Response to RC2**

Thank you for your time and constructive comments on the manuscript "Estimation of duration and its changes in Lagrangian observations relying on ice floes in the Arctic Ocean utilizing sea ice motion product". We will carefully consider all comments and make corresponding changes in our revised manuscript based on these suggestions.

The paper focuses on improving Arctic Lagrangian observations by analyzing long-term sea ice motion data (1979-2020). The study evaluates suitable deployment zones for ice camps and buoys by using sea ice motion products and incorporating atmospheric circulation patterns like the Arctic Oscillation and Arctic Dipole. The authors highlight the declining survival time of ice floes and the increasing challenges for Lagrangian observations due to climate change. This research is highly relevant, given the rapid transformations in Arctic sea ice dynamics and the growing need for precise observational data for climate modeling. By integrating trajectory simulations with EEZ constraints, the study provides actionable insights for future observational campaigns.

I appreciate the exhaustiveness of all sea ice thermodynamic and dynamic throughout the manuscripts. However, I have several concerns regarding the Area Of Interest (AOI) , the methodology protocol, and some logic explanations. Therefore, I recommend that the paper undergo major revisions before it can be considered for publication.

**General Comments:**

**1.** I realized that the aim of the work is to provide the reference for the ideal deployment locations in the central Arctic Ocean (in Line 73), but I don't understand why author choose the starting points region just within the rectangular area instead of within the EEZ boundary since EEZ anyway is divided into BG and IPD? So, I am not sure the motivation, is it just want to find the ideal deployment region within the rectangular only?

**Reply:** Indeed, the primary objective of our manuscript is to provide insights into the ideal deployment areas in the central Arctic Ocean for the buoys or ice camps to ensure that they drift in the central Arctic Ocean beyond the exclusive economic zones of various coastal countries for a sufficient amount of time.

To save search and recognition time for optimal deployment areas, we have defined a rectangular search area within the open sea in the current manuscript. We have also stated in the text that beyond this rectangular area, the deployments over other peripheral areas cannot meet the requirement of obtaining sufficient survival time of >= 9 months in the open sea area beyond the exclusive economic zones. In order to further confirm the reliability of our identification results, based on the suggestions of the reviewers, we will conduct a global search over the open sea area of the central Arctic Ocean using sea ice motion mean field from 1979 to 2023, which will be

extended in the revised version, to further demonstrate the rationality of the defined rectangular area.

**2**. Data and method part: How do you interpolate the 25 km ice motion when employing Lagrangian methods, linear or inverse distance weighting? Do you apply the Lagrangian method from start to end without any regridding during the period? How do the results compare to a semi-Lagrangian approach?

**Reply:** (1) We used the bilinear interpolation method to interpolate ice motion speeds; (2) The original grid of sea ice motion products is the Ease-Grid, and the grid of the study area is the Polar Stereographic Grid, so before applying the Lagrangian method, the original sea ice motion is regridded at the study area grid point, and then bilinear interpolation is used to obtain the Lagrangian sea ice motion speed; (3) We will consult the relevant literature and apply the semi-Lagrangian method for further validation in Section 4.1. Additionally, we will include a comparison of reconstruction results of ice trajectories derived from the semi-Lagrangian method with those obtained from the Lagrangian method.

**3**. The validation of buoy trajectories seems to focus on data after 2014. Are there additional buoy datasets available from earlier periods? If not, are the selected buoys representative and exhaustive for this study?

**Reply:** Due to the need to use buoy data that has not been assimilated into NSIDC's sea ice motion products, the range of available data is relatively limited. Following the suggestion, we will attempt to collect earlier available buoy data and extend the data time span to the earlier years. Currently, to ensure representativeness, we use 10 buoys for each region of BG and TPD. To further collect data, we will also consider the buoys in both regions to ensure that we have a relatively consistent number of validation samples for both regions.

**4**. Another interesting point to explore could be backtracking trajectories instead of forward tracking. For trajectories with >9 months survival time (ST), does the backtrack reveal that their starting points are mostly within the rectangular AOI? This may provide valuable insights into uncertainties and trajectory origins.

**Reply:** Thank you for this suggestion. We plan to integrate these insights into Section 4.1 to explore the uncertainty of sea ice trajectory and assess the effectiveness of recommended deployment areas. Using trajectories with a survival time ST exceeding 9 months, we will obtain the spatial distribution of the endpoints of these forward trajectories during the study years, extending to 1979-2023, and determine the main hotspot areas where the endpoints of the ice trajectories originating from recommended deployment areas are clustered by geographic models such as spatial clustering algorithms. Following this, we plan to reconstruct backward trajectories from the grid points within the hotspot region of endpoints, to investigate whether the terminations of these backward trajectories still can reach the recommended

deployment area. Then we can further evaluate the reliability of reconstructed sea ice drift trajectories using the data of sea ice motion field through this closed calibration evaluation method.

**5**. When using 2m air temperature for calculating Freezing Degree Days (FDD), how was the daily value derived - was it simply a mean of hourly data? Providing clarity on this calculation is crucial for reproducibility. How about the bias in ERA5 temperature.

**Reply:** We use the daily data obtained by averaging the hourly data of 2-m air temperature when calculating FDD, and we will use the daily data of ERA5 directly for comparison. We will also add some descriptions and previous verification results of the bias of the 2-m air temperature data of ERA5 in the Arctic region.

**6**. I am more interested in Figure 5, which is more pratically in the future. Shouldn't you further add more recommendation on the deployment for the future based on the 2007-2020 analysis (and also, could you longegate the time span from 1979-2023), and further make some uncertainties or high-recommend and mediate-recommend about the region? Since now for me, the all materials somehow distract me about the whole motivation. Incorporating uncertainty estimates and differentiating regions into high-recommendation and moderate-recommendation zones would greatly enhance the practical utility of the paper. As it stands, the extensive materials somewhat distract from the core motivation of the study.

**Reply:** According to this suggestion, we will extend the time span to 1979-2023. Based on this, we will subdivide the recommendation degree in all figures containing recommended deployment areas into moderate and high recommendation zones. Among them, the high recommendation area is determined based on the probability distribution of grid points. We will add discussions on uncertainty estimation and provide suggestions for future deployment based on the recommended zones.

7. Section 3.3, I'm not sure how much information related to the motivation can get obtained from here, please considering make them concrete.

**Reply:** Thanks for this comment. Due to the significant uncertainty in the data on sea ice thickness, the results of the impact of changes in sea ice thickness on future deployment recommendations or the operation of ice camps may not be reliable. Therefore, we will remove the section on sea ice thickness. For Section 3.3, since Section 3.2 discusses the thermodynamic impact of atmospheric forcing on sea ice, we will retain the content related to the ice-wind speed ratio and relocate it to Section 3.2 to discuss the dynamic impact of atmospheric forcing on sea ice. The changes in the response of sea ice to atmospheric forcing are of great significance for considering the near-year-round operation and maintenance of future ice camps, as well as for the interdisciplinary studies on the interactions between sea ice and lower atmosphere or upper ocean.

**8**. Section 4.1 requires further elaboration. In particular, I recommend adding an uncertainty analysis or sensitivity test to strengthen the robustness of the findings.

**Reply:** Thanks for this suggestion. To enhance the robustness of the results, we plan to add uncertainty analysis in Section 4.1. Specifically, to improve the representativeness of the verification results, we will try to collect more and earlier buoy data, with approximately consistent quantity of the buoys drifting over the BG and TPD regions. In addition, we will add discussions on methods, comparing the characteristics and applicability of the Lagrangian method and the semi-Lagrangian method and analyzing the uncertainty of the trajectory endpoint using the closed calibration evaluation method based on the further reconstructed backward trajectories.

**9**. I don't fully capture the Table 1 concerning its physical mechanism, first of all, how to understand the autumn CAI only have the obvious significant correlation with longitude in BH, but more correlated with both IPD and IPD/BH in latitude.

**Reply:** Table 1 showed the correlation between the atmospheric circulation indices and the longitude or latitude of the sea ice drift trajectory endpoint. Actually, CAI represents the air pressure gradient difference between the east and west of the central Arctic (94°N, 90°W, and 84°N, 90°E), which could characterize the intensity of TPD. In the BG region, sea ice motion is mainly driven by the anticyclonic circulation, so CAI mainly affects the longitude of the sea ice trajectory. In the TPD region, sea ice mainly advects meridionally, so CAI affects the latitude of the ice trajectory more significantly.

**Specific Comments:**

Line 23: change to "as the sea ice thins"
**Reply:** Thank you,we will revise it.

Line 117: use "optimal" instead of "most optimal"
**Reply:** Thanks for the suggestion, we will revise it.

Line 308-309: I am not sure about the statement since we don't know the casuality between
ice motion, wind circulation, near surface ocean current/stress. It is truly that sea ice motion, wind speed, ocean surface stress increase with climate change, but correlation doesn't give us some ideas in who is the trigger and who is the influencer. Could you provide more evidence.
**Reply:** Thanks, we also recognized this statement is not precise. We will check our result and revise this sentence to avoid doubts.

Line 347: "form" to "from"
**Reply:** We will revise this typing mistake.

Line 393-395, can you explain why?
**Reply:** We will consult relevant literature and add sentences to explain why the BH does not reveal more effective interpretability in the BG region.

---

## Author Response (AR1)

**Response to Editor**

Dear Dr. Xichen Li,

Thank you for giving us the opportunity to revise the manuscript. We sincerely appreciate the constructive comments and suggestions from you and reviewers, which have significantly improved the quality and clarity of this work. Please see below for a summary of our revision:

1) **Introduction:** We revised the introduction to better highlight the theme and significance of this study.

2) **Methodology:** To clarify the rationality of the study area selection, we conducted a search analysis for the potential preferred area over the entire central Arctic Ocean.

3) **Results and Discussion:** We subdivided the ideal deployment areas and added deployment recommendations, removed the assessment of sea ice thickness from section 3.3 and merged section 4.3 and 4.4 to present more reliable and clear results.

4) **Additional Analysis:** We have extended the study period, which is now from 1979–1980 to 2022–2023, and added new analyses of precipitation and snowfall along the trajectories. For validation, we expanded the buoy dataset, conducted a closed-loop test using backward trajectory analysis, and added comparisons between the results derived from semi-Lagrangian and Lagrangian methods to further evaluate the reliability of our reconstructed result.

Below, we provide point-by-point responses to the comments, line numbers refer to the revised manuscript with track changes.

Thank you again for your time and consideration.

Best regards,
Ruibo Lei, Xiaoping Pang and co-authors

**Response to RC1**

Thank you for your time and constructive comments on the manuscript "Estimation of duration and its changes in Lagrangian observations relying on ice floes in the Arctic Ocean utilizing sea ice motion product". We have considered comments carefully and incorporated practically all of them in the revised manuscript.

**Comments:**

This manuscript is interesting to me, in which the ideal deployment locations in the central Arctic Ocean for ice camp or buoy have been discussed by using SIM product, and the main standard is to ensure that Lagrangian observations can last a period as long as possible. The logic and structure of the manuscript are ok for me, but I still suggest the authors should address the following issues.

**(1)** L66, "The Arctic sea ice is mainly driven by wind and oceanic current stresses". Maybe you mean the drift of Arctic sea ice or sea ice dynamics is driven by….

**Reply:** Yes, we revised this sentence to "Arctic sea ice dynamics is mainly driven by wind and oceanic current stresses...". In the revised manuscript, as the introduction was revised to emphasize the main topic, we removed this sentence.

**(2)** L73-77 tell the motivation of this study. But I am not sure if there is any similar study on this topic? From the statement here, this study is the first one considering the ideal deployment location in Arctic Ocean.

**Reply:** In the introduction, we provided an overview and summary of the main achievements of previous studies on the ideal deployment location or logistical considerations in the Arctic Ocean (**lines 70-77**).

**(3)** The content in the introduction section is a little confusing. What exactly you want to summary in this part? Sea ice dynamics? Pervious ice stations? Or something else?

**Reply**: Thank you for pointing it out. Actually, we want to emphasize the history and challenges of long-term observations based on ice camps or buoys, as well as the importance of ideal deployment areas identification to ensure continuous and effective observations. We have reorganized and revised the introduction to improve the relevant expressions (**lines 65-67, 70-77**).

**(4)** L110. The definition of the rectangular area in Fig.1 is still somewhat arbitrary to me. Maybe you can put the EASE-Grid as the background and then select some from

all of them. "The reasons for this diagnosis will be given later." Please specify where you have discussed this problem.

**Reply:** Thanks to your suggestion. To demonstrate the rationality of the definition of the rectangular area, we conducted surveys for the potential preferred area over the entire central Arctic Ocean using sea ice motion mean field from 1979 to 2023. The results showed that 91.6% of starting points within the rectangular area allow ≥ 9 months of drift trajectories without drifting into the EEZ or beyond ice zone, exceeding other region within the central Arctic regions. Even in the last decade (2013–2023), with thinner and younger ice, the rectangular region had 25.6% of effective starting points, compared to an absence of such points in other regions. So it is extremely difficult to find suitable areas to deploy ice camps or buoys in the central Arctic Ocean outside of our defined rectangular region in order to maintain Lagrangian observations for a long enough period of time. The relevant result was added to Section 2.1 (**lines 117-128**).

**(5)** Section 2.3. If the ice floe is broken into pieces during drifting, is there any impact on the calculation of the survival time (ST)?

**Reply:** In reality, the breakup of an ice floe during drift certainly has an impact on survival time. However, in our calculations, we cannot be judged when and where the ice floes breakup would occur using the sea ice motion product. Such impact and challenges for maintaining observation of long time series using ice camps or buoys are not within the scope of this study. However, we still highlighted this impact in the conclusion section.

In this study, we mainly use sea ice concentration and exclusive economic zone boundaries to determine survival time of ice camps or buoys. We have already explained such limitations of this study in **lines 543-547, 550-552**. Based on our analysis and diagnosis, we wish to identify potential ideal areas for buoy or ice camp deployments. In actual operation, the operating time of buoys or ice camps also depends on the breakup or collapse of the ice camp or buoy and its supporting ice floe, the formation of melt ponds and the intrusion of polar bear, etc. Therefore, what we infer should be the maximum potential survival time. Nevertheless, we argue that such analysis is still necessary for the selection of deployment areas for buoys or ice camps, especially when we hope to obtain a sufficiently long time series of longer observations.

**(6)** L197, "FDD(TDD) refers to the integral of near-surface air temperatures below…". What kind of air temperature? hourly average or daily average?

**Reply:** We used the daily average air temperature for the integration of FDD and TDD, as explained in **line 226**.

**(7)** Section 3.2. Air forcing such as temperature and long-wave radiation have been investigated here. How about the precipitation? Snowfall poses an important impact on sea ice growth and decay.

**Reply:** Following this suggestion, we calculated the precipitation and snowfall along the trajectories reaching the BG and TPD regions, respectively, and added some discussions on their regional differences (**lines 302-309**).

**(8)** Section 3.3. The ice-wind speed ratio in Fig.8 is also overall lower than the typical values in free-drift analytical solution. You can also discuss this difference, maybe relating to sea ice concentration.

**Reply:** Thank you for the suggestion. We discussed why the ice-wind speed ratios are lower than typical values in free-drift analytical solutions in **lines 340-343**.

Thank you for your time and constructive comments on the manuscript "Estimation of duration and its changes in Lagrangian observations relying on ice floes in the Arctic Ocean utilizing sea ice motion product". We have carefully considered all comments and made corresponding changes in our revised manuscript based on these suggestions.

The paper focuses on improving Arctic Lagrangian observations by analyzing long-term sea ice motion data (1979-2020). The study evaluates suitable deployment zones for ice camps and buoys by using sea ice motion products and incorporating atmospheric circulation patterns like the Arctic Oscillation and Arctic Dipole. The authors highlight the declining survival time of ice floes and the increasing challenges for Lagrangian observations due to climate change. This research is highly relevant, given the rapid transformations in Arctic sea ice dynamics and the growing need for precise observational data for climate modeling. By integrating trajectory simulations with EEZ constraints, the study provides actionable insights for future observational campaigns.

I appreciate the exhaustiveness of all sea ice thermodynamic and dynamic throughout the manuscripts. However, I have several concerns regarding the Area Of Interest (AOI) , the methodology protocol, and some logic explanations. Therefore, I recommend that the paper undergo major revisions before it can be considered for publication.

**General Comments:**

**1.** I realized that the aim of the work is to provide the reference for the ideal deployment locations in the central Arctic Ocean (in Line 73), but I don't understand why author choose the starting points region just within the rectangular area instead of within the EEZ boundary since EEZ anyway is divided into BG and IPD? So, I am not sure the motivation, is it just want to find the ideal deployment region within the rectangular only?

**Reply:** Following this suggestion and the comments from other reviewer, we conducted surveys for the potential preferred area over the entire central Arctic Ocean using sea ice motion mean field from 1979 to 2023, and added a statement of the study area in the revised version to further demonstrate the rationality of our defined rectangular area (**lines 117-128**). Details are given below:

Using the 1979-2023 climatology field of sea ice motion, the ice trajectories were reconstructed for the period starting from October 1 to September 30, i.e., one

year, using a total of 4289 points over the entire central Arctic Ocean as starting points. With a threshold of the trajectory being in the central Arctic Ocean beyond the exclusive economic zones (EEZs) >= 9 months, we found that 91.6% of the defined rectangular area is eligible, while only 36.0% of the peripheral area is eligible. Even during 2013–2023, a period facing more severe challenge due to the thinner and younger ice, the rectangular region contained 25.6% of effective starting points, while the other regions had no such points. So it is extremely difficult to find suitable areas to deploy ice camps or buoys in the central Arctic Ocean outside of our defined rectangular area in order to maintain Lagrangian observations for a long enough period of time. This suggests that considering the computational cost, the definition of our defined rectangular area is reasonable.

**2**. Data and method part: How do you interpolate the 25 km ice motion when employing Lagrangian methods, linear or inverse distance weighting? Do you apply the Lagrangian method from start to end without any regridding during the period? How do the results compare to a semi-Lagrangian approach?

**Reply:** Thank you for suggestion. We have revised the relevant sentences and added the discussion. The details are as follows:

(1) We used the bilinear interpolation method to interpolate ice motion speeds (**lines 180-181**);

(2) The original grid of sea ice motion products is the EASE-Grid, and the grid of the study area is the polar stereographic grid, so before applying the Lagrangian method, the original sea ice motion is regridded at the study area grid point, and then bilinear interpolation is used to obtain the Lagrangian sea ice motion speed;

(3) We added a comparison of the semi-Lagrangian method with the Lagrangian method, and validated the ice trajectories reconstructed by the semi-Lagrangian method using buoy trajectories in Section 4.1. The results showed that the reconstructed trajectories from the semi-Lagrangian method are highly similar to the results from the Lagrangian method, however the results derived from the Lagrangian method have a relatively high accuracy compared to that obtained from the semi-Lagrangian method. (**lines 374-385**)

**3**. The validation of buoy trajectories seems to focus on data after 2014. Are there additional buoy datasets available from earlier periods? If not, are the selected buoys representative and exhaustive for this study?

**Reply:** As suggested, we collected earlier buoy data spanning from 2010 to 2023, obtaining the data from 15 buoys for each region of BG and TPD for validation, and

**Table A1** (**line 557**) shows the basic information of all 30 buoys. We revised the corresponding result in Section 4.1.

**4**. Another interesting point to explore could be backtracking trajectories instead of forward tracking. For trajectories with >9 months survival time (ST), does the backtrack reveal that their starting points are mostly within the rectangular AOI? This may provide valuable insights into uncertainties and trajectory origins.

**Reply:** Following the suggestion, we conducted a closed-loop examination by reconstructing backward ice drift trajectories from endpoints of the original forward-reconstructed trajectories. Using trajectories with a survival time (ST) exceeding 9 months, we obtained the spatial distribution of the endpoints of these forward trajectories during the study years, extending to 1979/1980–2022/2023. We then used the Density-based Clustering tool to identify the hotspot region of the endpoints. After identifying the main hotspot regions, we reconstructed backward trajectories from the grid points (795 points) within the hotspot region to check whether the terminations of the backward trajectories can reach the recommendation deployment areas. Results showed 66.3% of backward trajectories returned to recommended regions, which demonstrated the high confidence in the reconstruction method of ice trajectory. We added some text in **lines 386-393**, as well as **Figure 9** to describe this closed-loop examination to further evaluate the reliability of the reconstructed ice drift trajectories.

**5**. When using 2m air temperature for calculating Freezing Degree Days (FDD), how was the daily value derived - was it simply a mean of hourly data? Providing clarity on this calculation is crucial for reproducibility. How about the bias in ERA5 temperature.

**Reply:** We revised the description for the FDD/TDD calculations and added sentences in the data section describing the evaluation of the ERA5 variables used in this study (**lines 158-161, 227-228**). The details are as follows:

(1) For trajectories with a 90% probability of reaching the BG and TPD regions, the difference between the 2-m temperature calculated from ERA5 hourly data and the daily mean 2-m temperature obtained directly from ERA5 ranged from -2.66 × $10^{-4}$ K to 2.65 × $10^{-4}$ K, with an average difference of 7.18 × $10^{-7}$ K. This indicates that the discrepancy between them is negligible. Figure 1 below illustrates that the FDD and TDD of 2-m temperatures derived directly from daily ERA5 data and those calculated from hourly ERA5 data exhibit nearly identical trends from 1979 to 2022. Notably, the increasing trend of FDD in the BG and TPD regions is consistent, with both datasets showing equal values.

[Figure]

Figure 1. Comparison of FDD/TDD using daily 2-me temperature calculated from hourly ERA5 data versus daily 2-m temperature directly from ERA5.

(2) We consulted the literature on ERA5 temperature assessment in the Arctic. The correlation between ERA5 2-m temperature and N-ICE2015 observations is above 0.9 in January-May, but drops to 0.57 in June. In January-June, the 2-m temperature data exhibit a warm bias, with a bias range of about 0.8 to 3.4°C (Graham et al.,2019). Compared with the IMB and Snow Buoy observations, the ERA5 2-m air temperature also exhibits a warm bias ( < 4°C), with a larger warm bias in September-May and a smaller warm bias in June-August (Wang et al.,2019). Overall, the warm bias in ERA5 2-m temperature may lead to relatively small values in the FDD calculations, but our study mainly focuses on the difference in the trend of FDD in the BG and TPD regions. The influence of warm bias of 2-m temperature on the trend is relatively trivial compared to its absolute value.

Graham, R.M., Cohen, L., Ritzhaupt, N., Segger, B., Graversen, R.G., Rinke, A., Walden, V.P., Granskog, M.A. and Hudson, S.R.: Evaluation of Six Atmospheric Reanalyses over Arctic Sea Ice from Winter to Early Summer, Journal of Climate, 32(14): 4121-4143, https://doi.org/10.1175/JCLI-D-18-0643.1, 2019.

Wang, C., Graham, R.M., Wang, K., Gerland, S. and Granskog, M.A.: Comparison of ERA5 and ERA-Interim near-surface air temperature, snowfall and precipitation over Arctic sea ice: effects on sea ice thermodynamics and evolution, The Cryosphere, 13(6): 1661-1679, https://doi.org/10.5194/tc-13-1661-2019, 2019.

**6**. I am more interested in Figure 5, which is more pratically in the future. Shouldn't you further add more recommendation on the deployment for the future based on the 2007-2020 analysis (and also, could you longegate the time span from 1979-2023),

and further make some uncertainties or high-recommend and mediate-recommend about the region? Since now for me, the all materials somehow distract me about the whole motivation. Incorporating uncertainty estimates and differentiating regions into high-recommendation and moderate-recommendation zones would greatly enhance the practical utility of the paper. As it stands, the extensive materials somewhat distract from the core motivation of the study.

**Reply:** Based on this suggestion, we extended the study period, is now from 1979–1980 to 2022–2023 because the sea ice motion data for 2024 are not yet available.

As shown in Figure 2, the probability distribution of ST > 9 months ranges from 0.09 to 0.91 in the study region, so we classified the regions with probabilities between 0.75 and 0.85 as moderate-recommendation zones, and those greater than 0.85 as high-recommendation zones. On this basis, we subdivided the ideal deployment areas in **Figures 4, 5, and 11** into moderate- and high- recommendation zones in the revised manuscript for further discussion. Furthermore, we also added discussions on uncertainty estimation and provided suggestions for future deployment based on the recommendation zones (**lines 246-249, 269-272**).

[Figure]

Figure 2. Probability distribution of ST for no less than 9 months in the study region.

**7.** Section 3.3, I'm not sure how much information related to the motivation can get obtained from here, please considering make them concrete.

**Reply:** Due to the significant uncertainty in the data on sea ice thickness, the results of the impact of changes in sea ice thickness on future deployment recommendations or the operation of ice camps may not be reliable. Therefore, we removed the section on sea ice thickness and relocated the content related to the ice-wind speed ratio to Section 3.2 (**lines 333-350**).

**8.** Section 4.1 requires further elaboration. In particular, I recommend adding an uncertainty analysis or sensitivity test to strengthen the robustness of the findings.

**Reply:** Thank you for the suggestions. To improve the robustness of our results, we have added an uncertainty analysis in Section 4.1, consisting of the following three aspects:

1) To improve the representativeness of the validation results, we have collected buoy data obtained from earlier years from 2010 to 2023 (**line 557, Table A1**), thereby having 15 buoys for each region of BG and TPD for validation;

2) We added a discussion on the comparison of the reconstructed trajectories using the semi-Lagrangian and Lagrangian methods (**lines 374-385**);

And 3) As mentioned earlier, for trajectories with ST> 9 months, we reconstructed backward trajectories from endpoint hotspot region and analyzed the uncertainty of the endpoints of the trajectories using a closed calibration assessment method (**lines 386-393**).

**9**. I don't fully capture the Table 1 concerning its physical mechanism, first of all, how to understand the autumn CAI only have the obvious significant correlation with longitude in BH, but more correlated with both IPD and IPD/BH in latitude.

**Reply:** Table 1 showed the correlation between the atmospheric circulation indices and the longitude or latitude of the ice drift trajectory endpoint. Actually, CAI represents the air pressure gradient difference between the east and west of the central Arctic (94°N, 90°W, and 84°N, 90°E), which could characterize the intensity of TPD. In the BG region, sea ice motion is mainly driven by the anticyclonic circulation, so CAI mainly affects the longitude of the sea ice trajectory. In the TPD region, sea ice mainly advects meridionally, so CAI affects the latitude of the ice trajectory more significantly. We further improved the expression to make it clearer in the revised manuscript (**lines 407-408, 419-420**).

**Specific Comments:**

Line 23: change to "as the sea ice thins"

Line 117: use "optimal" instead of "most optimal"

Line 347: "form" to "from"

**Reply:** We have revised them (**lines 23, 131 and 355**) according the comments.

Line 308-309: I am not sure about the statement since we don't know the casuality between ice motion, wind circulation, near surface ocean current/stress. It is truly that sea ice motion, wind speed, ocean surface stress increase with climate change, but correlation doesn't give us some ideas in who is the trigger and who is the influencer. Could you provide more evidence.

**Reply:** Thanks, we removed this inaccurate statement and revised the relevant text (**lines 343-348**).

Line 393-395, can you explain why?

**Reply:** We added text to explain why the BH does not reveal more effective interpretability in the BG region (**lines 422-424**).

---

## Referee Report (RR1)

**Review of "Estimation of duration and its changes in Lagrangian observations relying on ice floes in the Arctic Ocean utilizing sea ice motion product"**

The manuscript presents an analysis of potential durations and changes in Lagrangian observation trajectories in the Arctic Ocean using sea ice motion (SIM) products from 1979 to 2020. The authors propose a reconstruction method to track synthetic buoys and analyze long-term trends in the survivability of sea ice-based platforms. The work is particularly relevant in the context of planning future Arctic field campaigns, where floe lifetime and trajectory uncertainty are critical.

The manuscript includes a substantial amount of data analysis, including validation against real buoys and exploration of relationships between trajectory duration and climate indices. However, many of the core scientific and methodological concerns raised in the first round have not been sufficiently addressed. Below, I detail unresolved issues, expanded requests, and additional suggestions for a more rigorous and impactful revision. I recommend **minor revisions**.

**General Comments:**

1. The manuscript continues to define synthetic deployment points within a fixed rectangular AOI in the central Arctic Ocean. This choice is not convincingly justified, especially given that the study emphasizes EEZ constraints and international boundary considerations later in the paper. Why not define initial locations based on EEZ boundaries or current common deployment areas (e.g., MOSAiC, N-ICE2015)? The reader is still left wondering whether the motivation is to assess *ideal* deployment zones within political constraints, or simply to map climatological drift patterns in a limited domain. These are fundamentally different objectives and should be clearly separated and addressed.

2. The manuscript still does not adequately explain how sea ice motion vectors are interpolated onto buoy positions during Lagrangian tracking. Is the interpolation linear? Bilinear? IDW? Are velocity fields regridded before or during integration? Moreover, the paper should comment on whether the use of a fully Lagrangian approach (i.e., step-by-step advection) could be compared to semi-Lagrangian methods, or if any correction is made to account for accumulated drift bias over long periods.Reliance on low-resolution, outdated SIM data undermines the validity of conclusions.

3. The study relies solely on the NSIDC Polar Pathfinder 25 km product, which is known to underestimate short-term variability and to smooth dynamic features relevant for station-keeping and observational fidelity. No comparison is made to higher-resolution products (e.g., OSI SAF 6.25 km, MEEREIS), nor are any bias assessments provided. Without this, conclusions about changes in survivability or trajectory characteristics over time are difficult to evaluate.

4. Trajectory validation continues to rely on a small number of buoys, mostly post-2014. The use of earlier IABP buoys or additional campaign data (e.g., SHEBA, DAMOCLES) could significantly strengthen the credibility of results. Moreover, distance between reconstructed and real trajectories is reported, but no error metrics such as RMSE, angular deviation, or trajectory similarity (e.g., Fréchet distance) are provided. This is especially important since the method is used to infer survivability duration over multi-month periods. Also, backtracking analysis of long-duration (>9 months) trajectories would provide insight into which ice origins produce the most stable paths. This could help refine AOI definitions or identify zones of persistent ice retention. Unfortunately, this suggestion remains unaddressed, despite being a straightforward addition that could substantially improve the paper's utility.

5. Section 3.3 remains somewhat abstract and disconnected from the core analysis. The discussion of drift trends and sea ice circulation patterns is mostly descriptive and lacks linkage to either the survivability calculations or campaign implications. Consider condensing or more tightly integrating this section with the trajectory duration results.

6. Section 4.1: no sensitivity or uncertainty analysis. The survivability results are presented without any robustness testing. How sensitive are results to the time of year, small changes in starting position, or minor variations in drift vector? Even a basic bootstrapping or ensemble test would help confirm that the observed patterns are not artifacts of the interpolation method or limited starting conditions.

7. Table 1: interpretation of CAI correlations still vague. The correlation analysis between duration and atmospheric circulation indices is repeated from the original version, with no added clarity. For example, why does autumn CAI correlate with longitude in BH but not IPD? What does the correlation in latitude for both suggest about the spatial mode of influence? Without physical interpretation, the correlation analysis is not actionable and feels disconnected from the rest of the study.

**Specific Comments:**

Eq. (1-3): Clarify the interpolation technique and its potential accumulation of drift error over time. Describe how errors in SIM data are propagated in trajectory reconstructions. Include justification for using 25 km resolution and for excluding any higher-res or recent products.

Table/Figure 3: Add quantitative error metrics. Distance alone is insufficient, what are direction errors, turning biases, etc.? Expand comparison beyond one buoy trajectory (e.g., include MOSAiC or additional IABP floats).

Why limit the results fixed the time span between 1979-2020 instead of 1979-2024?

---

## Author Response (AR3)

**Response to Editor**

Dear Dr. Xichen Li,

Thank you for giving us the opportunity to revise the manuscript. We are grateful to reviewer 2 for the valuable comments that helped to improve the quality of this work. Please see below for a summary of our revision:

1) **Methodology:** We clarified the rationality of the study area selection, explained the interpolation technique and its potential accumulation of drift error over time, and the reasons for choosing the 25km sea ice motion product instead of high-resolution product.

2) **Results and Discussion:** We condensed Section 3.2, revised the backtracking analysis in Section 4.1, and revised the interpretation of the impact of the atmospheric circulation indices in Section 4.2 to present clearer results.

3) **Additional Analysis:** For validation, we added the SHEBA and DAMOCLES campaign buoy data and quantitative error metrics to assess the reliability of our reconstructed result, and the effects of year, starting position and interpolation method on the duration to test the robustness of the duration results.

Below, we provide point-by-point responses to the comments, line numbers refer to the revised manuscript with track changes.

Thank you again for your time and consideration.

Best regards,
Ruibo Lei, Xiaoping Pang and co-authors

**Response to RC2**

Thank you for your time and constructive comments on the manuscript "Estimation of duration and its changes in Lagrangian observations relying on ice floes in the Arctic Ocean utilizing sea ice motion product". We have carefully reviewed all comments and addressed the key concerns in our revised manuscript based on these suggestions.

The manuscript presents an analysis of potential durations and changes in Lagrangian observation trajectories in the Arctic Ocean using sea ice motion (SIM) products from 1979 to 2020. The authors propose a reconstruction method to track synthetic buoys and analyze long-term trends in the survivability of sea ice-based platforms. The work is particularly relevant in the context of planning future Arctic field campaigns, where floe lifetime and trajectory uncertainty are critical.

The manuscript includes a substantial amount of data analysis, including validation against real buoys and exploration of relationships between trajectory duration and climate indices. However, many of the core scientific and methodological concerns raised in the first round have not been sufficiently addressed. Below, I detail unresolved issues, expanded requests, and additional suggestions for a more rigorous and impactful revision. I recommend minor revisions.

**General Comments:**

**1.** The manuscript continues to define synthetic deployment points within a fixed rectangular AOI in the central Arctic Ocean. This choice is not convincingly justified, especially given that the study emphasizes EEZ constraints and international boundary considerations later in the paper. Why not define initial locations based on EEZ boundaries or current common deployment areas (e.g., MOSAiC, N-ICE2015)? The reader is still left wondering whether the motivation is to assess ideal deployment zones within political constraints, or simply to map climatological drift patterns in a limited domain. These are fundamentally different objectives and should be clearly separated and addressed.

**Reply:** We fully understand the need for this choice of AOI to be fully justified. The decision to use a rectangular area is based on the following considerations: in the central Arctic Ocean, i.e., the high Arctic that excluded the exclusive economic zones (EEZs), it is computationally intensive and inefficient to carry out trajectory reconstruction for 1979-2023 for all 4289 grid points. To effectively reduce the amount of computation while maintaining representativeness, a rectangular region was selected as the study area. Tests were conducted to demonstrate the representativeness of the rectangular study area. The details are as follows:

Using the 1979-2023 climatology field of sea ice motion, the ice trajectories were reconstructed for the period starting from October 1 to September 30, i.e., one year, using a total of 4289 points over the entire central Arctic Ocean as starting points. With a threshold of the trajectory being in the central Arctic Ocean beyond the EEZs >= 9 months, we found that 91.6% of the defined rectangular area is eligible, while only 36.0% of the peripheral area is eligible. Even during 2013–2023, a period facing more severe challenge due to the thinner and younger ice, the rectangular region contained 25.6% of effective starting points, while the other regions had no such points. So it is extremely difficult to find suitable areas to deploy ice camps or buoys in the central Arctic Ocean outside of our defined rectangular area in order to maintain Lagrangian observations for a long enough period of time.(**lines 119-128**)

This demonstrates that our defined rectangular area, while balancing computational efficiency, effectively represents the sea ice dynamics in the central Arctic Ocean. The primary objective of this study is to assess ideal deployment strategies under constraints

**2**. The manuscript still does not adequately explain how sea ice motion vectors are interpolated onto buoy positions during Lagrangian tracking. Is the interpolation linear? Bilinear? IDW? Are velocity fields regridded before or during integration? Moreover, the paper should comment on whether the use of a fully Lagrangian approach (i.e., step-by-step advection) could be compared to semi-Lagrangian methods, or if any correction is made to account for accumulated drift bias over long periods.Reliance on low-resolution, outdated SIM data undermines the validity of conclusions.

**Reply:** Thank you for suggestion. We added description about the interpolation method during Lagrangian tracking and emphasized comparison between the Lagrangian method and semi-Lagrangian method. The details are as follows:

1) In the Lagrangian tracking process, we used the bilinear interpolation to interpolate sea ice motion vectors onto buoy positions. The velocity field is dynamically interpolated at each integration step. This approach preserves instantaneous spatial variability but may smooth small-scale features due to the 25 km grid resolution. We added descriptions in **lines 180-182**;

2) We have added a comparison of the semi-Lagrangian method with the Lagrangian method in Section 4.1 last time and have now revised the text. The comparison results showed that the reconstructed trajectories from the semi-Lagrangian method are highly similar to the results from the Lagrangian method, however using the buoy trajectory as validation data, we found that the results derived from the Lagrangian method have a relatively high accuracy compared to that obtained from the semi-Lagrangian method. (**lines 372-384**).

And 3) We added the explanation that accumulated drift bias over long periods in the reconstruction process is small so no correction was done (**lines 182-187**). For the NSIDC SIM product used in this study, although there are errors in the individual motion estimates, these errors do not accumulate over long term tracking because the motion estimates are largely unbiased (Tschudi et al.,2020).This is further supported by the study of Tschudi et al. (2010), who found a drift error of 27 km over 293 days of tracking(with bilinear interpolation), which suggests that errors can still be kept within limits over long term tracking.

Tschudi, M. A., Fowler, C., Maslanik, J. A., and Stroeve, J.C., 2010. Tracking the movement and changing surface characteristics of Arctic sea ice. IEEE J. Sel. Topics Appl. Earth Observ. Remote Sens., 3, 536–540, https://doi.org/10.1109/JSTARS.2010.2048305.

Tschudi, M.A., Meier, W.N., and Stewart, J.S., 2020. An enhancement to sea ice motion and age products at the National Snow and Ice Data Center (NSIDC). Cryosphere, 14(5), 1519-1536, https://doi.org/10.5194/tc-14-1519-2020.

**3.** The study relies solely on the NSIDC Polar Pathfinder 25 km product, which is known to underestimate short-term variability and to smooth dynamic features relevant for station-keeping and observational fidelity. No comparison is made to higher-resolution products (e.g., OSI SAF 6.25 km, MEEREIS), nor are any bias assessments provided. Without this, conclusions about changes in survivability or trajectory characteristics over time are difficult to evaluate.

**Reply:** Thank you for comment. The choice of the NSIDC 25 km sea ice motion product was primarily motivated by its long-term consistency (1979–present) and availability for large-scale trend analysis, which is essential for our climatological analysis. While higher-resolution products (e.g., OSI SAF 6.25 km or MEEREIS) better resolve short-term variability, their limited temporal coverage (typically post-2010) preclude their use for assessing long-term trends.

We acknowledge that the NSIDC product's smoothing of small-scale dynamics may affect trajectory details, and Section 2.3 discussed this limitation, adding that the errors accumulated in trajectory reconstruction from SIM data are relatively small (**lines 180-187**). Future work will integrate higher-resolution products to refine the representation of short-term processes.

**4.** Trajectory validation continues to rely on a small number of buoys, mostly post-2014. The use of earlier IABP buoys or additional campaign data (e.g., SHEBA, DAMOCLES) could significantly strengthen the credibility of results. Moreover, distance between reconstructed and real trajectories is reported, but no error metrics such as RMSE, angular deviation, or trajectory similarity (e.g., Fréchet distance) are provided. This is especially important since the method is used to infer survivability

duration over multi-month periods. Also, backtracking analysis of long-duration (>9 months) trajectories would provide insight into which ice origins produce the most stable paths. This could help refine AOI definitions or identify zones of persistent ice retention. Unfortunately, this suggestion remains unaddressed, despite being a straightforward addition that could substantially improve the paper's utility.

**Reply:** Thank you for the suggestions. To improve the robustness of our results, we have revised the following three aspects in Section 4.1:

1) We have collected buoy data obtained from 1997 to 2023 and added the SHEBA and DAMOCLES campaign data for validation (**line 150 and Table A1**);

2) We added error metrics such as the RMSE and Fréchet distance between the buoy and reconstructed trajectories to improve the confidence of the validation (**lines 218-222 and Section 4.1**);

And 3) Actually, we added an analysis of backward trajectories last time and have now revised the text to emphasize this backtracking analysis (**lines 385-393**). For trajectories with ST > 9 months, we reconstructed backward trajectories from hotspot region and displayed the endpoints distribution (gray dots) in Fig. 9b. 66.3% of the endpoints of the backward trajectories were able to return to the moderate- and high- recommendation zones (the region surrounded by the blue or red line in Fig. 9b), indicating that the recommended zones, as a source area for sea ice, have stable paths.

**5.** Section 3.3 remains somewhat abstract and disconnected from the core analysis. The discussion of drift trends and sea ice circulation patterns is mostly descriptive and lacks linkage to either the survivability calculations or campaign implications. Consider condensing or more tightly integrating this section with the trajectory duration results.

**Reply:** We have deleted Section 3.3 and moved the ice response to wind forcing to Section 3.2, and condensed Section 3.2 (**lines 298-346**).

**6.** Section 4.1: no sensitivity or uncertainty analysis. The survivability results are presented without any robustness testing. How sensitive are results to the time of year, small changes in starting position, or minor variations in drift vector? Even a basic bootstrapping or ensemble test would help confirm that the observed patterns are not artifacts of the interpolation method or limited starting conditions.

**Reply:** Thanks for the suggestion. We added some text in Section 4.1, **lines 394-402**. To test the robustness of the survival results, we added the effects of year, starting position and interpolation method on the duration, using the starting points reaching the BG or TPD region over 90% as an example. The statistical results

showed that the standard deviation of duration in 1979-2022 amounted to 76.0 days. It is noteworthy that with the significant acceleration of sea ice motion in recent years (Sumata et al., 2023), this standard deviation decreases to 20.7 days for the period 2007-2022, which corresponds to 7.2% of the mean duration. It should be noted that this study focuses on the mean duration characteristics at the climate scale. The difference between the mean duration in case of starting position offset (±10 km) and the results obtained for the starting position accounts for about 1.1%, and the mean duration obtained from the ice trajectories reconstructed with the nearest neighbor interpolation differs from the results obtained with the bilinear interpolation by an average of 7.6%, which proves that small changes in the initial coordinates and the interpolation method barely affect the reliability of the duration.

Sumata, H., de Steur, L., Divine, D.V., Granskog, M.A., and Gerland, S., 2023. Regime shift in Arctic Ocean sea ice thickness. Nature, 615(7952), 443-449, https://doi.org/10.1038/s41586-022-05686-x.

**7.** Table 1: interpretation of CAI correlations still vague. The correlation analysis between duration and atmospheric circulation indices is repeated from the original version, with no added clarity. For example, why does autumn CAI correlate with longitude in BH but not IPD? What does the correlation in latitude for both suggest about the spatial mode of influence? Without physical interpretation, the correlation analysis is not actionable and feels disconnected from the rest of the study.

**Reply:** We revised the relevant physical interpretation to add clarity of the correlation analysis in Section 4.2 (**lines 415-420, 421-422 and 429-432**).

**Specific Comments:**

Eq. (1-3): Clarify the interpolation technique and its potential accumulation of drift error over time. Describe how errors in SIM data are propagated in trajectory reconstructions. Include justification for using 25 km resolution and for excluding any higher-res or recent products.

**Reply:** We added some text to explain the interpolation technique and the fact that potential accumulation of drift error over time is small (**lines 180-187**), and emphasized the rationale for choosing the 25-km product in the data section (**lines 137-140**).

Table/Figure 3: Add quantitative error metrics. Distance alone is insufficient, what are direction errors, turning biases, etc.? Expand comparison beyond one buoy trajectory (e.g., include MOSAiC or additional IABP floats).

**Reply:** Thanks, we added RMSE and Fréchet distance as metrics of quantitative error, and we revised the descriptions to emphasize that the actual comparison is over the range of buoy trajectories in Table A1 (**Section 4.1**).

Why limit the results fixed the time span between 1979-2020 instead of 1979-2024?

**Reply:** Since the sea ice motion data for 2024 are not yet available, our study period is now from 1979–1980 to 2022–2023.